# Efferent feedback controls bilateral auditory spontaneous activity

Yixiang Wang [1], Maya Sanghvi[1], Alexandra Gribizis[1,2], Yueyi Zhang[1], Lei Song[3,4], Barbara Morley [5], Daniel G. Barson [1], Joseph Santos-Sacchi[1,3,6], Dhasakumar Navaratnam[1,3,7] & Michael Crair [1,8 ✉]

In the developing auditory system, spontaneous activity generated in the cochleae propagates into the central nervous system to promote circuit formation. The effects of peripheral firing patterns on spontaneous activity in the central auditory system are not well understood. Here, we describe wide-spread bilateral coupling of spontaneous activity that coincides with the period of transient efferent modulation of inner hair cells from the brainstem medial olivocochlear system. Knocking out α9/α10 nicotinic acetylcholine receptors, a requisite part of the efferent pathway, profoundly reduces bilateral correlations. Pharmacological and chemogenetic experiments confirm that the efferent system is necessary for normal bilateral coupling. Moreover, auditory sensitivity at hearing onset is reduced in the absence of pre-hearing efferent modulation. Together, these results demonstrate how afferent and efferent pathways collectively shape spontaneous activity patterns and reveal the important role of efferents in coordinating bilateral spontaneous activity and the emergence of functional responses during the prehearing period.

[1] Department of Neuroscience, Yale University School of Medicine, New Haven, CT, USA. [2] Max Planck Florida Institute for Neuroscience, One Max Planck Way, Jupiter, FL, USA. [3] Department of Surgery (Otolaryngology), Yale University School of Medicine, New Haven, CT, USA. [4] Department of Otolaryngology-Head and Neck Surgery, Shanghai Ninth People's Hospital, Shanghai Jiao Tong University School of Medicine, Shanghai, China. [5] Center for Sensory Neuroscience, Boys Town National Research Hospital, Omaha, NE, USA. [6] Department of Cellular and Molecular Physiology, Yale University School of Medicine, New Haven, CT, USA. [7] Department of Neurology, Yale University School of Medicine, New Haven, CT, USA. [8] Kavli Institute for Neuroscience, Yale University, New Haven, CT, USA. ✉email: michael.crair@yale.edu

Developing sensory systems generate distinct patterns of spontaneous activity to facilitate self-organization and circuit formation before the onset of sensory experience[1–4]. In the cochleae, groups of inner hair cells (IHC) fire bursts of action potentials that propagate to central auditory nuclei via ascending pathways to coordinate activity throughout the auditory system prior to hearing onset[5–8]. Although central auditory activity may inherit many properties directly from the periphery, it manifests with distinct patterns and novel features[8–10]. Consequently, understanding the characteristics of spontaneous activity in the central auditory system may provide insight into the maturation of circuit connectivity and the development of higher-order auditory functions.

During the prehearing period, patterns of spontaneous firing of IHCs change across development and vary by tonotopic position in the immature cochlea[11,12], feeding variable ascending inputs to central circuits. Furthermore, during this time period, IHCs' activity is modulated by transient efferent feedback from medial-olivocochlear (MOC) neurons[13–16], which depends on α9/α10 nAChRs and coupled short-conductance potassium (SK2) channels[17]. High variability in the peripheral inputs, combined with high fidelity of local coordinated firing, provide substrates for activity-dependent Hebbian plasticity that can promote maturation of precise spatial maps[2,18]. However, exactly how ascending and descending pathways interact and shape activity patterns to instruct circuit formation remain largely unexplored.

Recent studies have shown coordinated spontaneous activity in the inferior colliculi (IC) at P6–P8, utilizing SNAP25-GCaMP6s mice with wide-field epifluorescent microscopy[8,10]. The calcium transients, detected with the GCaMP sensor, manifest as band-shaped bursts that align across the expected future tonotopic axis in the IC. Here, we conducted a systematic study of spatiotemporal and correlational properties of in vivo spontaneous activity in the IC over the entire prehearing period, ranging from postnatal day 0 to 13 (P0–P13). Our results revealed a changing profile of in vivo spontaneous activity and evolving bilateral connectivity in the auditory system that peaks early in the prehearing period and slowly declines prior to hearing onset. Intriguingly, the strength of bilateral coupling paralleled the time course of a transient cholinergic modulation imposed on IHCs by the MOC efferent system[15,17,19]. Mice lacking the α9/α10 nicotinic acetylcholine receptor (nAChR), a necessary constituent of efferent modulation[20–22], displayed severely disrupted bilateral correlations. Chemogenetic and pharmacological experiments in vivo used to acutely manipulate pre- and post-synaptic components of the efferent circuits, indicate that the MOC system was necessary for normal bilateral coupling. Finally, we observed a significant elevation of auditory thresholds at hearing onset in α9/α10 nAChR knockout mice. Together, our results indicate a profound influence of the MOC system, a descending efferent circuit, in coordinating ascending tonotopic bilateral spontaneous activity throughout the auditory system and promoting normal auditory circuit development.

## Results

### Evolving spontaneous activity across the prehearing period.
We first examined how spatiotemporal properties of in vivo spontaneous activity changes from birth (P0) until hearing onset (~P13) using wide-field calcium imaging in the mouse IC at six different ages: P0–P1, P3–P4, P6–P7, P9–P10, P11–P12, and P13 (Fig. 1a, b). The IC contains three major divisions: the central nucleus, the lateral cortex (LCIC), and the dorsal cortex (DCIC). Only the dorsal parts of the IC, including DCIC and LCIC, are accessible for calcium imaging[8,23]. Consistent with the previous reports[8,10], typical spontaneous calcium transients in the dorsal surface of the IC appear as stationary and discrete events that (1) have a relatively homogeneous intensity profile along their major axis, (2) show confined spread along the future tonotopic axis that is approximately perpendicular to the major axis (Fig. 1c and Supplementary Fig. 1b, major axes labeled as magenta arrows), and (3) appear as a single band in the most medial part or dual bands in more lateral parts of the IC (see also Supplementary Movie S1). We refer to these calcium transients as "spontaneous bands" or "spontaneous events". We developed a suite of analysis tools for automatic and quantitative descriptions of the spatiotemporal properties of the spontaneous bands (see "Methods"). We specifically characterized the activity level and spatial profile of spontaneous bands across different ages (Fig. 1d–f). Spontaneous bands were present immediately after birth (P0–P1), though with relatively small peak amplitude, and low event frequency. This is consistent with the immature cochlear machinery at P0–P1[11]. At P3–P4, bands appeared with larger peak amplitude and higher event frequency but had wider spans along the future tonotopic axis compared to bands observed at later stages (Fig. 1f). At P6–P7, event frequency continued to increase, indicating that spontaneous activity reached a more active state, and the highest event frequency was recorded at P11–P12, right before hearing onset. We also observed that inter-peak intervals (IPI) decreased monotonically from P0 to P12 (Supplementary Fig. 1d), consistent with the monotonic increase of event frequency (Fig. 1e). Note that spontaneous events at P11–P12 also manifested with more confined width (Fig. 1f) and shorter average duration (Supplementary Fig. 1e) compared to earlier ages, suggesting that spontaneous bands were more concentrated both spatially and temporally. The spontaneous activity started to diminish at P13, as peak amplitude and event frequency dropped significantly, signaling the end of the internally generated activity and the beginning of external inputs (Fig. 1d, e). Taken together, the spontaneous activity becomes more frequent and spatially refined from birth until hearing onset in vivo.

We also conducted dimensionality reduction analysis on the calcium data (Supplementary Fig. 1i–k). We resolved functional modules that are consistent with the known characteristic organization of the IC solely based on spontaneous activity patterns. Specifically, diffusion map results displayed two mirror-symmetrical domains (Supplementary Fig. 1k), matching the tonotopic-reversal structure in the dorsal IC[23]. These results suggest that functional features of the IC are embedded in the intrinsic dynamics of spontaneous activity even before hearing onset.

### Bilateral coupling diminishes before hearing onset.
Given that central auditory circuits are highly bilateral, we probed correlative features of spontaneous bands in both hemispheres of the IC by examining global bilateral correlations (Fig. 1g, j) and local (seed-based) correlations (Fig. 1h, i, k) in the calcium imaging data. After controlling for the activity that was external to the selected regions of interests (ROIs; all correlation maps in the study are based on partial correlations unless stated otherwise, see also "Methods"), background correlations within the IC or between the IC and the visual superior colliculus were reduced, while the salient correlation patterns were preserved (Supplementary Fig. 2a, see also "Methods"). We noticed that global bilateral correlations at P11–12 were significantly lower than at earlier ages (Fig. 1j), indicating that average neural activity became less coupled between the two hemispheres at this stage.

The global correlation analysis relies on measurements of activity averaged over the entire IC hemisphere and does not reflect the spatial profile of bilateral correlations or resolve location-specific information. Thus, we also examined spatial correlation patterns in the IC with seed-based correlation maps.

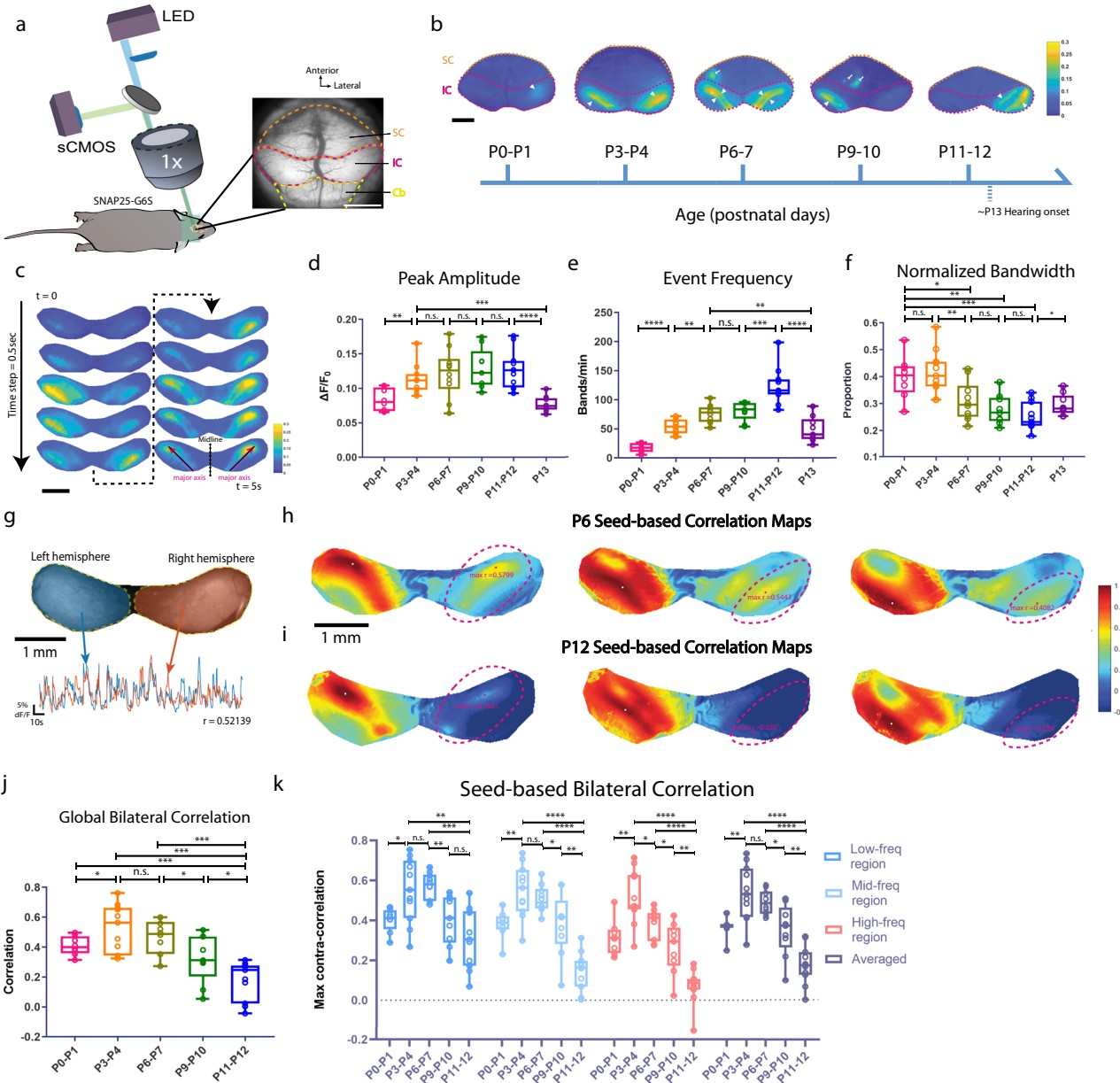

**Fig. 1 Spatiotemporal and correlational properties of prehearing spontaneous activity in the inferior colliculus. a** Experimental setup for wide-field calcium imaging showing a typical field of view in mice. Superior colliculi (SC), inferior colliculi (IC), and part of the cerebellum (Cb) are delineated in dashed orange, magenta, yellow lines, respectively. All movies are acquired at 10 Hz. **b** Example $\Delta F/F_0$ images showing spontaneous events at different postnatal ages. Superior colliculi (SC), inferior colliculi (IC) are delineated in dashed orange and magenta lines, respectively. White arrowheads: spontaneous bands in the IC. White arrows: spontaneous (retinal) waves in the SC. Colormap: parula (MATLAB). **c** Example montages ($\Delta F/F_0$) of spontaneous activity bands in the IC. Two magenta arrows represent major axes of the intensity profile in the left and right hemispheres. **d** Average peak amplitude of spontaneous activity bands across age groups. Defined as the mean $\Delta F/F_0$ amplitude at peak. **e** Event frequency across age groups. Quantified as the number of individual peaks (bands) per minute as identified in the line-scan analysis. **f** Average normalized bandwidth across age groups. Defined as the mean spatial half-width of the fluorescence peaks normalized by the width of the IC. **g** Schematics for analysis of global bilateral correlations. The left hemisphere and its corresponding mean fluorescent trace are colored in blue. The right hemisphere and its mean fluorescent trace are colored in orange. A partial correlation between the two traces was computed while regressing out the effect of the mean activity overall pixels outside the IC ($r = 0.52139$). **h** Example seed-based correlation maps at P6. Dashed magenta lines delineate symmetrical regions of interest (ROIs) in the contralateral hemisphere with respect to the reference seeds. Magenta dots denote maximum correlation in the ROIs (max $r$). White dots in the left hemisphere indicates seed locations. **i** Example seed-based correlation maps at P12. **j** Average global bilateral correlation across age groups (P0–P12). **k** Summary quantification of seed-based bilateral correlation grouped by future frequency regions. "Low", "mid", and "high" corresponds to maps where reference seeds located in the putative low-, mid-, and high-frequency regions. "Averaged" correlation is defined as the mean correlation averaged over the three regions. Postnatal ages are shown on the x-axis. Box plots in Fig. 1: hinges: 25 percentile (top), 75 percentile (bottom). Box whiskers (bars): Max value (top), Min value (bottom). The line in the middle of the box is plotted at the median. Significance marks: n.s. $p > 0.05$, *$p < 0.05$, **$p < 0.01$, ***$p < 0.001$, ****$p < 0.0001$, two-tailed unpaired t test with Welch's correction. A number of animals: P0–P1 ($N = 8$); P3–P4 ($N = 11$); P6–P7 ($N = 10$); P9–P10 ($N = 9$); P11–P12 ($N = 11$); P13 ($N = 9$). Scale bar indicates 1 mm. Source data and exact p values are provided as a Source Data file.

We used a high-throughput pipeline to generate numerous maps in parallel, among which we selected representative maps for reference seeds located in regions corresponding to future low-, mid-, and high- frequencies (Supplementary Fig. 2c). We noticed symmetrical correlation patterns in both hemispheres at P6 (Fig. 1h). Specifically, when the representative seed was in the medial IC (putative future low-frequency region), we observed a single correlation band on each side. When the representative seed moved laterally towards the future mid- or high-frequency region of the IC, we observed dual correlation bands that shifted correspondingly to more lateral regions on each side (see Supplementary Movie S2), reflecting the known reversal of tonotopic organization in each hemisphere[23]. Symmetrical patterns on the two sides of the IC indicate that coupling of bilateral spontaneous activity is itself tonotopic. At P12, however, correlation patterns waned substantially in the contralateral (with respect to the seeds) hemisphere (Fig. 1I). We used a novel graphical-user-interface (Supplementary Fig. 2b) to quantify maximum correlations within symmetrical regions of the contralateral hemisphere. Summary statistics in Fig. 1k shows that seed-based bilateral correlations (referred as SbBC hereafter) increase significantly between P0–P1 and P3–P4 and plateau through P6–P7. Noticeable drops occur at later ages, especially at P11–P12, where SbBCs across regions are drastically lower than the level of P3–P7. This increase-peak-decrease trend of bilateral correlations is indicative of developmental changes in functional connectivity between the two sides of the auditory system before hearing onset, which is unexpected given that the anatomy of the mature auditory circuit is highly bilateral. In addition, we noticed a tonotopic difference in bilateral correlations using SbBCs, with low-frequency regions exhibiting stronger correlations than those of high-frequency regions at P6–7 (Supplementary Fig. 2d), suggesting that coupling strengths can vary by tonotopic location.

The spatial profiles of the correlation patterns also provide insight into the local connectivity shaping the spatial properties of the spontaneous activity bands. We observed that the area of the correlations decreased, and the eccentricity of the correlation patterns increased over-development (Supplementary Fig. 2c–e), consistent with decreasing spontaneous activity bandwidths described in Fig. 1f. This suggests that the local connectivity becomes more spatially restricted and laminar-like as animals mature.

In summary, SbBC showed a trend similar to global bilateral correlations, but with more pronounced changes over time (Fig. 1j, k). Spontaneous activity in the two hemispheres of the IC has a confined local profile and becomes more uncoupled across different tonotopic regions when maturing towards hearing onset.

**α9/α10 nAChRs are required for normal bilateral coupling**. Why does bilateral coupling collapse while activity levels peak at P11–P12? We hypothesize that the MOC efferent system might be involved because MOC neurons transiently modulate IHCs during the pre-hearing period and are important for normal spontaneous firing patterns[24]. Previous studies showed that the presence of efferent cholinergic synapses, expression of relevant molecular machinery, and IHC responsiveness to acetylcholine all follow a similar temporal profile to the developmental timeline of bilateral correlations we describe here[14,15]. Moreover, MOC feedback circuits are known to project to both cochleae. Functionally, ipsilateral sounds can induce both ipsi- and contralateral MOC reflexes in mature animals[25,26].

MOC efferent axons modulate hair cells through the α9/α10 nicotinic acetylcholine receptors (α9/α10 nAChRs), which are coupled to hyperpolarizing potassium channels/SK2 channels[17] (see also Fig. 2a). In the nervous system, expression of α9/α10 nAChRs is limited to sensory hair cells[27]. We therefore examined

bilateral coupling of spontaneous activity in constitutive α9/α10 knockout mice on the same genetic background[22] as the "wild-type" SNAP25-G6s mice. At P6–P7, we saw similar bilateral correlation patterns in the α9/α10 double heterozygous animals (Fig. 2b) as in wild-type mice (Fig. 1j). However, knocking out α9, α10, or both subunits remarkably impaired correlation patterns in the contralateral hemispheres no matter where the reference seeds were located (Fig. 2b, c, and Supplementary Movie S3). Note that we grouped the single knockouts (α9 KO/α10 Het and α9 Het/ α10 KO) and the double knockouts (α9 KO/α10 KO) into the same group (α9/α10 KO) as we observed no significant difference in terms of correlation (Supplementary Fig. 3i) or activity levels (Supplementary Fig 3g, h) despite double knockouts showing a trend of more severe bilateral correlation defects (Supplementary Fig. 3i). We observed mild but significant increases in peak amplitude and event frequency (Fig. 2d, e), and similar event duration (Supplementary Fig. 3j) in knockout mice compared to their double-heterozygous littermates, suggesting that gross activity levels are not compromised in the knockouts. However, we did observe a profound decrease in bilateral correlations in α9/ α10 KOs at P6–P7 (Fig. 2f, g), reminiscent of typical P11–P12 patterns (Fig. 1k). Reduced bilateral correlations in the knockouts were also observed at P3–P4 (Supplementary Fig. 3a–f), reflecting the early onset of cholinergic modulation.

We then used a larger field of view to image bilateral IC and A1 simultaneously to examine whether bilateral correlations are similarly altered in the developing auditory cortex (Fig. 2j). Using seed-based correlation maps, we saw strong correlations between ipsilateral IC/ ipsilateral A1, ipsilateral IC/ contralateral A1, ipsilateral IC/ contralateral IC, and ipsilateral A1/ contralateral A1 in the control (Fig. 2h, k). In the knockout, however, strong correlations were only present between ipsilateral IC and ipsilateral A1 (Fig. 2i, k).

In summary, we found that α9/α10 nAChRs are required for strong bilateral coupling of spontaneous activity in the IC and auditory cortex. Bilateral correlation patterns were greatly diminished in the absence of α9/α10 nAChRs, as reflected by a significant reduction of correlation values in knockout animals compared to heterozygous littermates in both the brainstem and cortex. This phenotype was evident at least from P3 to P4 (Supplementary Fig. 3) through P6–P7 (Fig. 2), when the correlation level peaked in control mice (Fig. 1k). These results indicate that the MOC system plays an important role in coupling bilateral spontaneous activity before hearing onset.

**Olivocochlear neurons mediate bilateral correlations**. To investigate whether the loss of bilateral coupling requires acute cholinergic modulation, we directly manipulated cochlear function via topical pharmacological application to the round window (Fig. 3a, see "Methods"). We measured spontaneous activity in control animals (SNAP25-G6s) before and after application of the SK2 channel blocker apamin or the α9/α10 nAChR blocker alpha-Bungarotoxin to both cochleae (see also the schematics in Fig. 2a). After application of apamin, correlation patterns in the contralateral IC were largely abolished (Fig. 3e), similar to what we observed in the α9/α10 knockout. We compared global and seed-based correlations as well as activity levels before/after pharmacological application (see also Supplementary Movie S4). We noticed no change in event amplitude or frequency after drug application (Figs. 3b, c), while bilateral correlations dropped dramatically in the same animals (Fig. 3d, f). We next measured bilateral correlations in the IC and A1 in the same animals. Blocking SK2 channel abolished bilateral correlations while the correlation between ipsilateral IC and A1 was spared (Fig. 3g, h), analogous to the phenotypes observed in the α9/α10 knockout (Fig. 3h–k). We also observed a similar effect with the α9/α10 nAChR

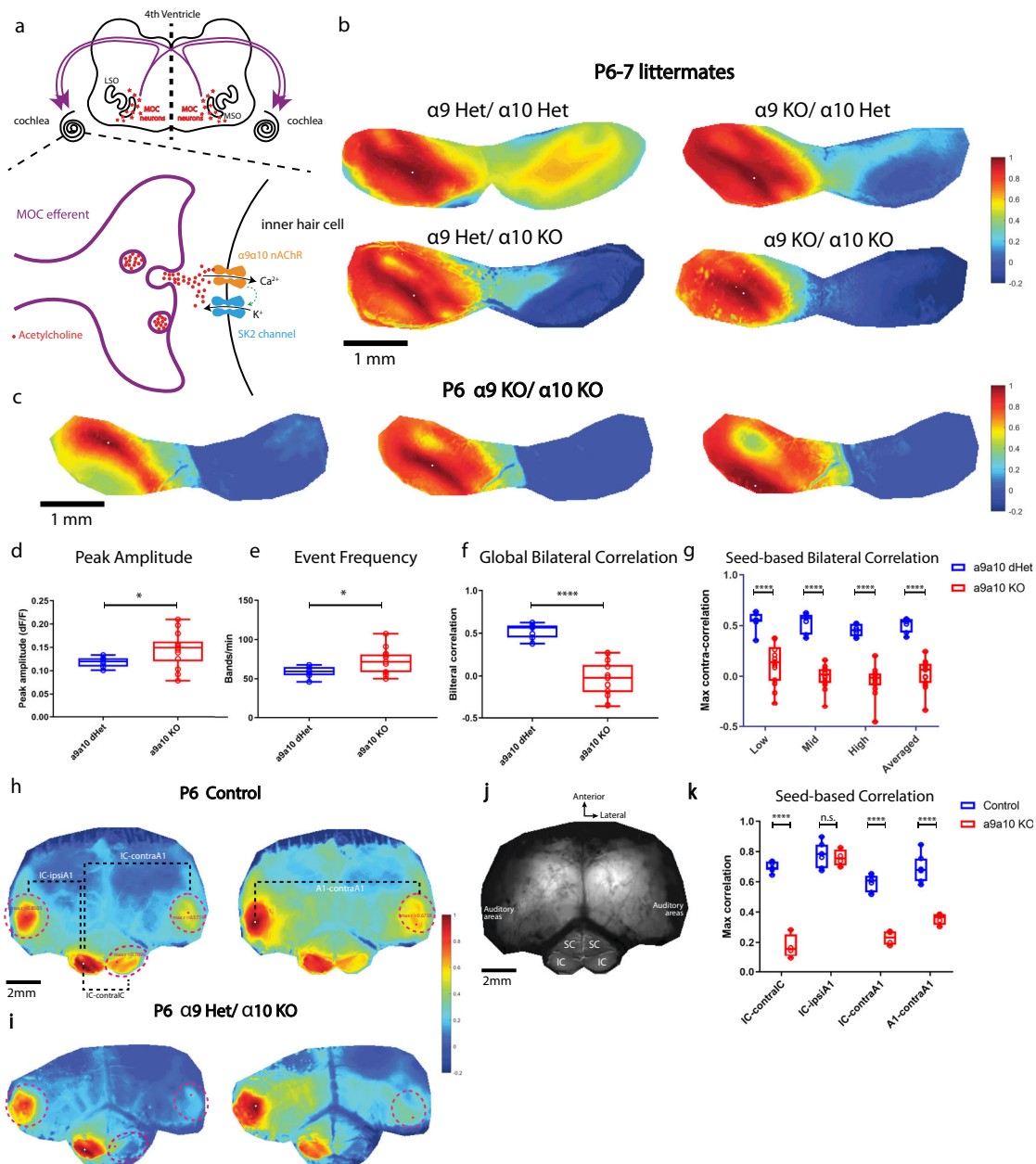

**Fig. 2 α9/α10 nAChR knockouts lack bilateral coupling of spontaneous activity in IC and A1. a** Top panel: Schematics of the medial-olivocochlear (MOC) efferent circuits. LSO lateral superior olive, MSO medial superior olive. Purple curved arrows: cochleae receiving bilateral MOC feedback. Bottom panel: Simplified schematics of a transient synapse between a MOC efferent axon and an inner hair cell. α9/α10 nAChR: alpha9/alpha10 nicotinic acetylcholine receptor. SK2 channel: KCNN2 potassium channel coupled with the α9/α10 nAChR. **b** Example correlation maps showing correlation patterns in the IC among littermates of different genotypes at P6–P7. Het heterozygous, KO knockout. Seeds (white dots) are all located in similar regions of the left IC. **c** Example IC correlation maps from a P6 α9/α10 double knockout animal. Three typical representative seeds (white dots) located in the future low-, mid-, and high-frequency regions. **d** Average peak amplitude. Similar to Fig. 1d. **d–g** a9a10 dHet: α9/α10 double heterozygous ($N = 7$); a9a10 KO: include single/double knockout of either or both α9/α10 subunits ($N = 14$). **e** Event frequency. Similar to Fig. 1e. **f** Global bilateral correlation. Similar to Fig. 1j. **g** Seed-based bilateral correlation. Similar to Fig. 1k. **h** Example correlation maps with a large field of view including both midbrain and cortex from a P6 SNAP25-G6s animal. Dashed magenta lines: ROIs defined to quantify maximum regional correlations. Left panel: Reference seed is in the left IC (white dot). IC-ipsiA1: Correlation between the ipsilateral IC (w.r.t the seed) and the ipsilateral A1. IC-contraA1: Correlation between ipsilateral IC and contralateral A1. IC-contraIC: Correlation between ipsilateral IC and contralateral IC hemispheres. Right panel: Reference seed is in the left A1 (white dot). A1-contraA1: Correlation between the ipsilateral A1 where the seed locates and the contralateral A1. A number of control animals (SNAP25-G6s) = 7. **i** Example correlation maps with a large field of view including both midbrain and cortex from a P6 α9 Het α10 KO animal. A number of α9/α10 KO animals = 4. **j** Field of view (grayscale mean fluorescent image) over the cortex and the midbrain. SC superior colliculi, IC inferior colliculi. Scale bar indicates 2 mm. **k** Seed-based correlation. IC-contraIC: Correlation between ipsilateral and contralateral IC. IC-ipsiA1: Correlation between ipsilateral IC and ipsilateral A1. IC-contraA1: Correlation between ipsilateral IC and contralateral A1. A1-contraA1: Correlation between ipsilateral A1 and contralateral A1. Box plot symbols similar to (**g**). n.s. $p > 0.05$, *$p < 0.05$, **$p < 0.01$, ***$p < 0.001$, ****$p < 0.0001$, two-tailed unpaired $t$ test with Welch's correction. **d** $p = 0.0228$, **e** $p = 0.0133$, **f** ****$p < 0.0001$, **g** ****$p < 0.0001$, **k** n.s. $p = 0.69495$, ****$p < 0.0001$. Source data and exact $p$ values are provided as a Source Data file.

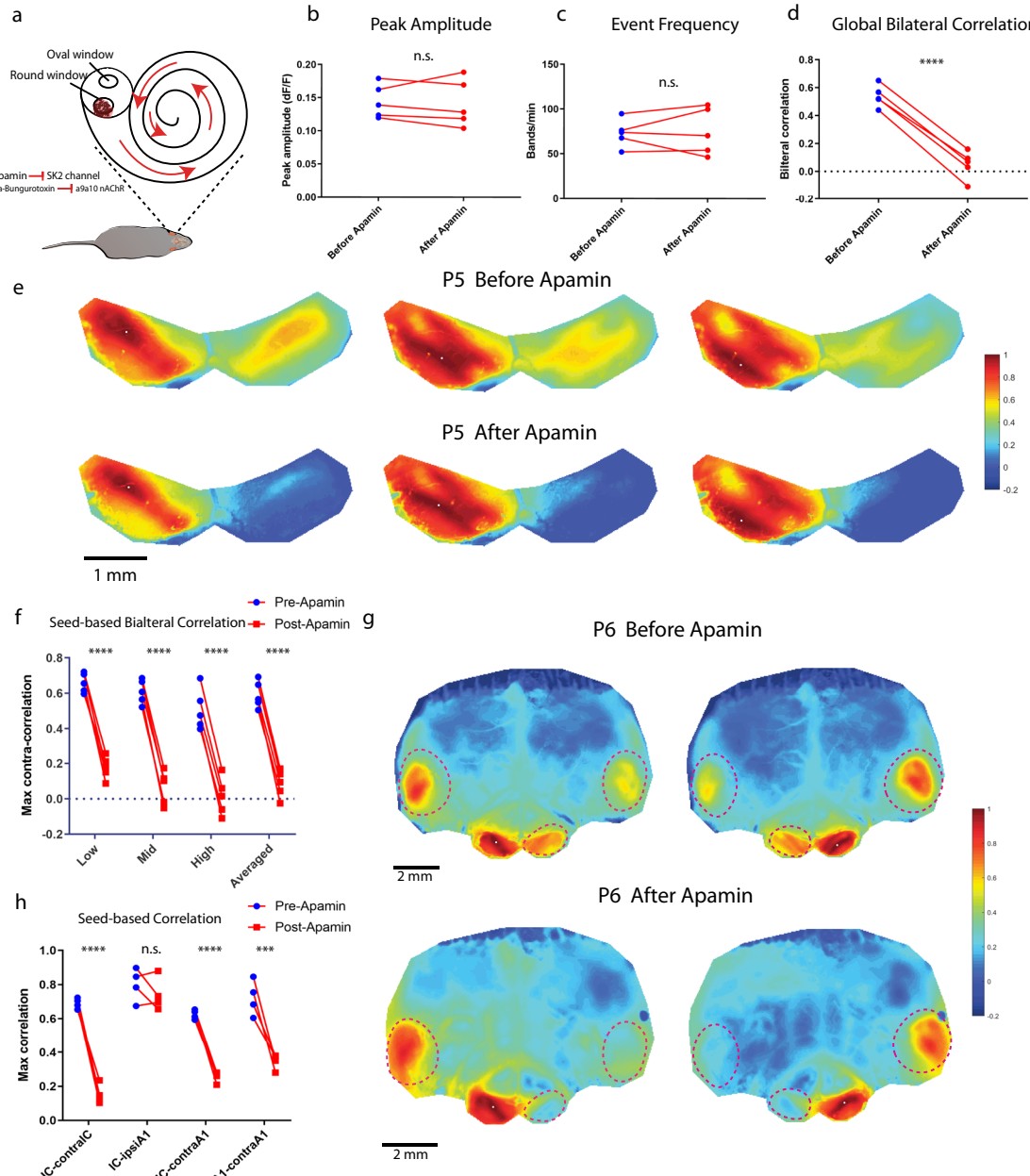

**Fig. 3 Acute apamin application abolishes bilateral coupling in vivo. a** Schematic showing in vivo pharmacology via delivery through the round window. **b** Example correlation maps showing correlation patterns in the IC before and after apamin application from the same SNAP25-G6s control animal at P5. Top panels: before apamin. Bottom panels: after apamin. Three typical representative seeds (white dots) located in the future low-, mid-, and high-frequency regions. **c** Average peak amplitude. Similar to Fig. 1d. **d** Event frequency. Similar to Fig. 1e. **e** Global bilateral correlation. Similar to Fig. 1j. **f** Seed-based bilateral correlation. Similar to Fig. 1k. **b–f** Number of animals (SNAP25-G6s) = 5. Scale bar denotes 1 mm. **g** Example correlation maps with a large field of view including both midbrain and cortex before and after apamin from the same P6 SNAP25-G6s animal. Top panels: before applying apamin. Bottom panels: after applying apamin. The white dot denotes the locations of seeds. Dashed magenta lines: ROIs defined to quantify maximum regional correlations. Similar to Fig. 2 h. **h** Seed-based correlation. IC-contraIC: Correlation between ipsilateral and contralateral IC. IC-ipsiA1: Correlation between ipsilateral IC and A1. IC-contraA1: Correlation between ipsilateral IC and contralateral A1. A1-contraA1: Correlation between ipsilateral A1 and contralateral A1. Significance marks: $^{n.s.}p > 0.05$, $^{****}p < 0.0001$, two-tailed unpaired $t$ test with Welch's correction. Number of animals (SNAP25-G6s) = 4. $^{n.s.}p > 0.05$, $^*p < 0.05$, $^{**}p < 0.01$, $^{***}p < 0.001$, $^{****}p < 0.0001$, two-tailed unpaired $t$ test with Welch's correction. **b** $p = 0.8810$, **c** $p = 0.8809$, **d** $^{****}p < 0.0001$, **f** $^{****}p < 0.0001$, **h** $^{n.s.}p = 0.417432$, $^{***}p = 0.000535$, $^{****}p < 0.0001$. Source data and exact $p$ values are provided as a Source Data file.

antagonist alpha-Bungarotoxin (Supplementary Fig. 4a–e). Applying saline with otherwise identical surgical procedures altered neither correlation profiles nor activity levels (Supplementary Fig. 4f–j). In summary, our results show that bilateral coupling of spontaneous activity is strongly influenced by cholinergic modulation, presumably from the MOC system, as blocking either SK2

channels or α9/α10 nAChRs in the cochlea acutely suppressed bilateral coupling in vivo.

To directly probe olivocochlear neurons' role in coupling bilateral spontaneous activity, we took advantage of cell-type-specific chemogenetics (Fig. 4a, g). We used designer receptors exclusively activated by designer drugs (DREADD)-based chemogenetic

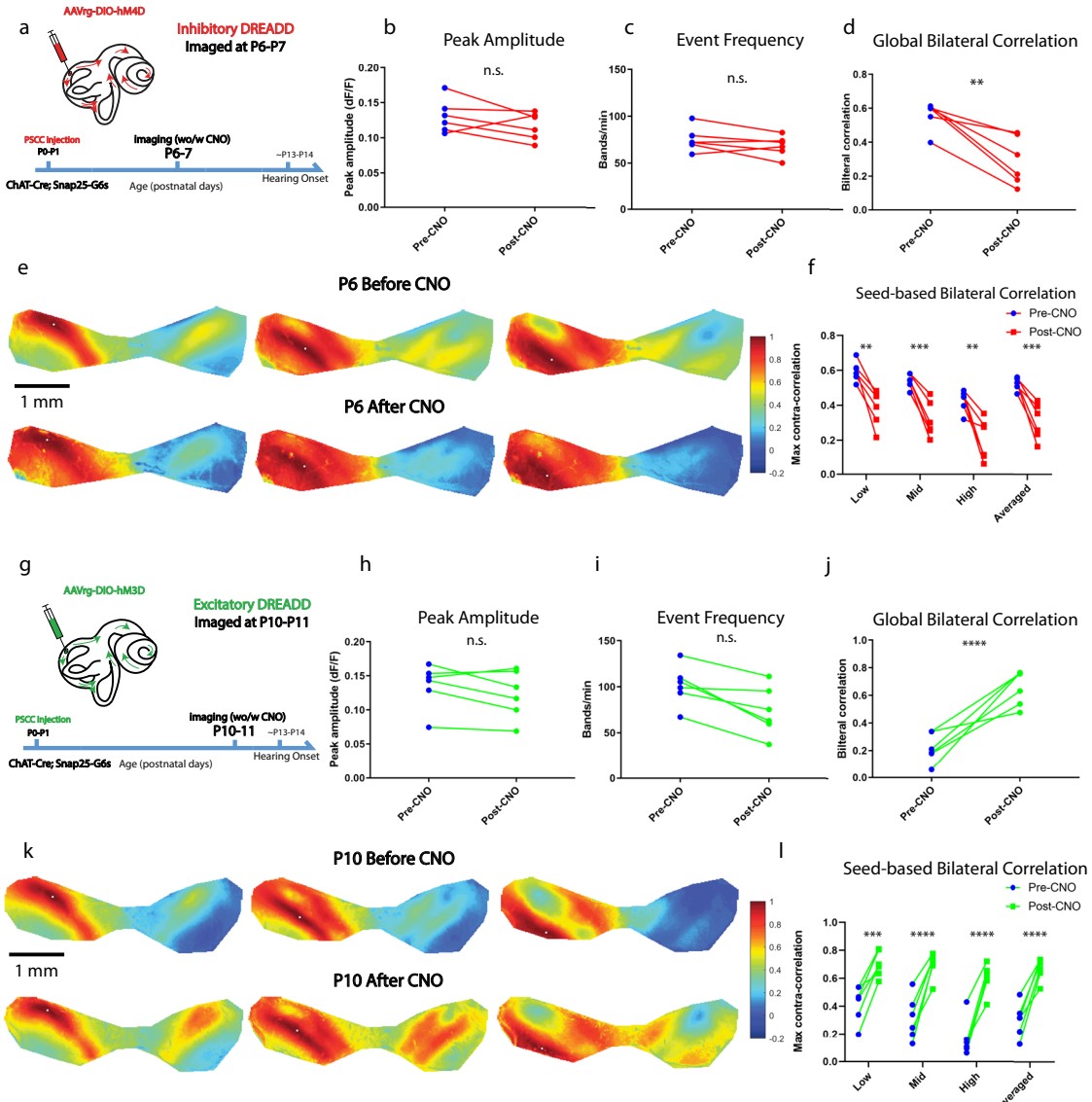

**Fig. 4 Chemogenetic manipulations can suppress or enhance bilateral coupling in vivo. a** Schematic showing post semicircular canal (PSCC) injection in ChAT-Cre;Snap25-G6s animals and the timeline of inhibitory chemogenetic experiments. CNO binding to hM4Di receptors (inhibitory muscarinic $M_4$ receptor-based Gi-coupled DREADD) expressed using a retrograde virus in olivocochlear neurons leads to inhibition of cochlear efferents. **b** Average peak amplitude. Similar to Fig. 1d. **c** Event frequency. Similar to Fig. 1e. **d** Global bilateral correlation. Similar to Fig. 1j. **e** Example correlation maps showing correlation patterns in the IC before and after CNO injection from the same animal at P6. Top panels: before CNO. Bottom panels: after CNO. Three typical representative seeds (white dots) located in the future low-, mid-, and high-frequency regions. **f** Seed-based bilateral correlation. Similar to Fig. 1k. **g** Similar to (**a**) but imaged at P10–P11 for excitatory chemogenetic experiments. CNO binding to retrograde virus-expressed hM3Dq receptors (excitatory muscarinic $M_3$ receptor-based Gq-coupled DREADD) in olivocochlear neurons leads to efferent neuron excitation. **h** Similar to (**b**) for excitatory chemogenetic experiments. **i** Similar to (**c**) for excitatory chemogenetic experiments. j Similar to (**d**) for excitatory chemogenetic experiments. **k** Similar to (**e**) but with an example animal at P10 for excitatory chemogenetic experiments. **l** Similar to (**f**) for excitatory chemogenetic experiments. Inhibitory DREADD experiments: number of animals = 6; theme color: Red. All imaged at P6–P7. Excitatory DREADD experiments: number of animals = 6; theme color: Green. All imaged at P10–P11. $^{n.s.}p > 0.05$, $^*p < 0.05$, $^{**}p < 0.01$, $^{***}p < 0.001$, $^{****}p < 0.0001$, two-tailed unpaired $t$ test with Welch's correction. **b** $p = 0.2921$, **c** $p = 0.3409$, **d** $p = 0.0037$, **f** $p$ values of Low, Mid, High, Averaged: 0.001369, 0.000527, 0.001991, 0.000549, **h** $p = 0.5201$, **i** $p = 0.0771$, **j** $^{****}p < 0.0001$, **k** $p$ values of Low, Mid, High, Averaged: 0.001644, 0.000412, 0.000094, 0.000098. Source data and exact $p$ values are provided as a Source Data file.

tools, which allow suppression or activation of specific neural populations upon application of DREADD agonizts[28]. Specifically, we injected rAAV-DIO-hM4D-mCherry (inhibitory DREADD) or rAAV-DIO-hM3D-mCherry (excitatory DREADD) retrograde viruses bilaterally in ChAT-CRE;SNAP25-G6s animals via posterior-semicircular-canal (PSCC) injections at P0–P1 (see "Methods"). Immunohistochemistry confirmed that this approach specifically targeted a subset of ChAT-positive olivocochlear neurons

(Supplementary Fig. 5a). We also verified that the cochlea received input from both contralateral and ipsilateral olivocochlear neurons with single-side injections (Supplementary Fig. 5b, c).

We first aimed to suppress bilateral coupling with Clozapine N-oxide (CNO), a DREADD agonist. We imaged the animals at P6–P7 when baseline correlations were highest (Fig. 1k). Peak amplitude, event frequency (Fig. 4b, c) and mean event duration (Supplementary Fig. 5d) did not significantly change

after CNO injection, while global and seed-based bilateral correlations reduced significantly (Fig. 4d–f, see also Supplementary Movie S5). Correlation patterns in the contralateral hemisphere were also impaired (Fig. 4e) as in the α9/α10 knockout or pharmacology experiments (Figs. 2 and 3). CNO alone in animals without virus injection did not affect bilateral correlations (Supplementary Fig. 5e–h). These results indicate that olivocochlear neurons are necessary for normal bilateral coupling in vivo.

We then aimed to "boost" the bilateral coupling with CNO. We imaged animals at P10–11 when baseline correlations were low (Fig. 1k). Although some MOC-IHC synapses start to degrade at this age[17], MOC mediated modulation can still be observed in IHCs in vitro[29]. We, therefore, subjected the olivocochlear neurons to chemogenetic hyper-activation with excitatory DREADD to enhance efferent modulation. We observed no significant change in activity levels after CNO injection (Fig. 4h–i), however, both global and seed-based bilateral correlations increased to peak levels (Fig. 4j–l. See also Supplementary Movie S5). Correlation bands in the contralateral hemisphere were restored after this manipulation, recapitulating the symmetrical patterns observed at P3–P7 (Fig. 4k). These results show that olivocochlear neurons are sufficient to induce bilateral coupling in vivo.

In summary, we demonstrated that bilateral coupling can be enhanced or degraded by suppressing or activating olivocochlear neurons via chemogenetic manipulation. Combined with the data from α9/α10 knockouts and pharmacological experiments, our results indicate that the MOC efferent system can induce and powerfully modulate bilateral coupling of pre-hearing spontaneous activity.

**Auditory thresholds are elevated in α9/α10 knockouts at the hearing onset.** To investigate whether auditory perception was affected in the α9/α10 knockout animals, we first measured auditory thresholds using the auditory brainstem response (ABR). Only ABR waves I–II were visible at P14.0 and were thus used to identify auditory thresholds (Fig. 5a, see also "Methods"). As ABR waves I–II reflect the collective firing of the eighth cranial nerve and cochlear nuclei (CN), responses in higher auditory nuclei downstream to the CN are difficult to detect with the ABR method. ABR thresholds were consistent with previous reports[30]. The difference between thresholds in α9/α10 knockout and control animals was insignificant despite a trend at P14.0 (Fig. 5b, two-way ANOVA, $F = 1.409$, $p = 0.0604$ on the column/genotype factor).

To access auditory responses in higher auditory nuclei, we measured auditory thresholds based on activity in the IC using wide-field calcium imaging at P14.0 (Fig. 5d, see also "Methods"). Robust calcium transients in response to acoustic stimuli were present in the IC across the frequency spectrum (Fig. 5d and Supplementary Movie S6). A tonotopic organization of isofrequency bands was observed in both control and knockout animals (columns enclosed in vertical dashed rectangles in Fig. 5d), with the medial region of the IC preferentially responding to low frequency, while more lateral regions favored higher frequencies. Typical responses consisted of pairs of bands symmetrically located against the tilted anterior-lateral axes (Supplementary Movie S6), consistent with the known tonotopic-reversal organization in the IC[23]. Interestingly, we observed a significant elevation on the threshold level of the α9/α10 knockout across the frequency spectrum at P14.0 compared to controls (Fig. 5c). The average threshold difference between controls and knockouts (across all frequencies) was around 20 dB. Taken together, our results indicate that auditory sensitivity in the α9/α10 knockout is impaired

at hearing onset despite minimal changes in auditory brainstem responses.

## Discussion

In vitro studies suggest a system of complex developmental regulation of auditory spontaneous activity in the cochleae[11]. We aimed to examine the developmental profile of pre-hearing spontaneous activity in central auditory circuits in vivo. Our experiments revealed the trajectories of key spatiotemporal features of IC spontaneous activity bands throughout the entire prehearing period. We observed characteristic organizational features embedded in the intrinsic dynamics of spontaneous activity in the prehearing IC, such as the tonotopic-reversals[23]. The large field-of-view afforded by wide-field imaging and the intact in vivo preparation permits the direct measurement of functional connectivity across ipsilateral and contralateral hemispheres that are millimeters apart. We observed an unexpected pattern of increase-peak-decrease in bilateral correlations of spontaneous activity between the two IC hemispheres across development. This indicates there are profound changes in bilateral connectivity over the 2-week time window before hearing onset (Fig. 1). This developmental timeline suggests that the MOC efferent system might play an important role in coupling bilateral spontaneous activity. Observations in AChR (α9/α10 nAChR) knockout mice, along with pharmacological and chemogenetic experiments (Figs. 2–4), confirmed that the MOC system is critical for enforcing normal bilateral coupling before hearing onset. These experiments reveal a novel role of MOC efferent modulation on auditory spontaneous activity. Finally, we also observed elevated auditory thresholds in the α9/α10 nAChR knockout mice immediately after hearing onset, based on calcium responses in the IC (Fig. 5). Our results, combined with the existing literature[24,31,32], demonstrate that disrupting spontaneous activity patterns undermines the normal development of auditory circuits.

Descending MOC efferent systems have been studied for decades[26,33–35]. In mature animals, outer hair cells (OHC) receive cholinergic feedback from MOC neurons. The MOC system suppresses the cochlear amplifier and mediates important functions, such as enhancing signal detection and sound localization in noisy backgrounds, protecting cochlear machinery from loud sounds, mitigating hidden hearing loss, and more[36]. Before hearing onset, however, MOC efferent fibers transiently synapse with IHC and modulate IHCs' spontaneous firing[13–16]. In both cases, this cholinergic modulation depends on α9/α10 nAChRs and coupled SK2 channels[17]. Previous studies show that animals lacking α9 nAChRs have altered temporal patterns of spontaneous firing[24]. In this study, we revealed an entirely new aspect of the MOC modulation in patterning spontaneous activity. By taking advantage of the large spatial scale of wide-field calcium imaging, we discovered the MOC's role in coupling bilateral spontaneous activity throughout the auditory system (Figs. 1–4).

These results have several implications. First, bilateral coupling of spontaneous activity may directly instruct bilateral circuit refinement. Central auditory circuits are heavily bilateral at almost all levels. During the prehearing period, peripherally generated spontaneous firing drives patterned activity in the entire auditory system and presumably promotes circuit maturation throughout. However, supporting cells and hair cells in both cochleae cannot communicate directly. Unlike sound waves, which are by nature correlated inputs of the same frequency components, pre-hearing spontaneous activity is independently initiated in the two cochleae. Without the MOC-mediated mechanism that we identified, uncoupled streams of peripheral activity might decrease synchronous activity between

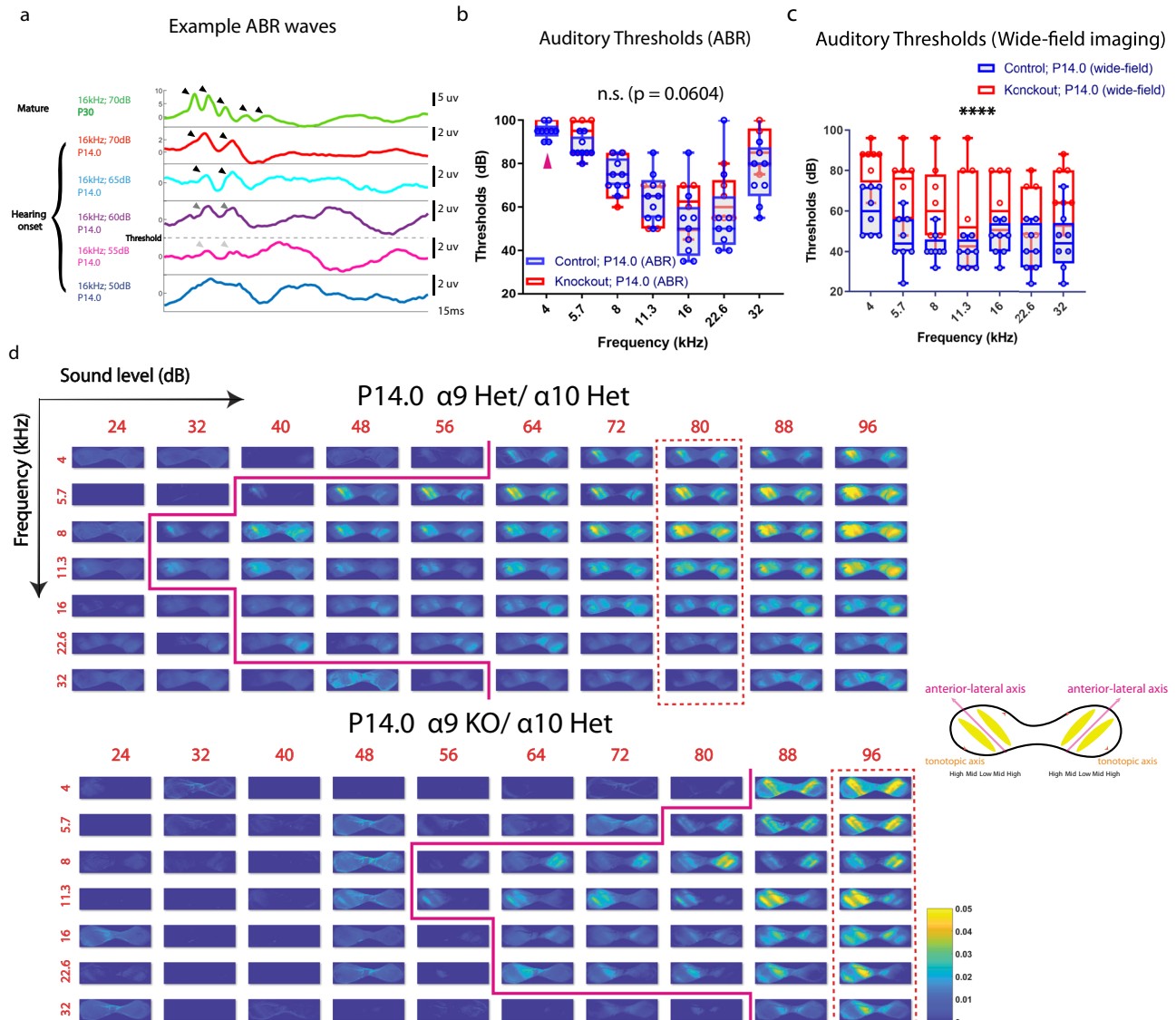

**Fig. 5 Elevation of auditory thresholds in α9/α10 nAChR knockouts at the hearing onset. a** Snapshots of example auditory brainstem response (ABR) curves. White arrowheads indicate ABR waves. Scale bar indicates 2 μV. The White dashed line represents the detection threshold. **b** Auditory thresholds measured by auditory brainstem response (ABR) in control and α9/α10 nAChR knockout animals at P14.0. A number of animals: Control P14.0 = 9 (GCaMP6s negative littermates of animals used in (**e**)). α9/α10 knockout P14.0 = 6. Magenta arrowheads indicate that part of the data is not available in the knockout group (two animals did not respond to 4 kHz tones.). n.s., two-way ANOVA, $F = 1.409$, $p = 0.0604$ on the column (genotype) factor. **c** Auditory thresholds measured by wide-field imaging. A number of animals: Control = 8 (wide-type SNAP25-G6s = 6; α9/α10 double heterozygous SNAP25-G6s = 2). Knockout = 8 (mix of single knockout of either α9 or α10 subunit and double knockout). Two-way ANOVA, $F = 31.98$, ****$p < 0.0001$ on the column (genotype) factor. **d** Example $\Delta F/F0$ response images to different acoustic stimuli. Row direction: Sound level. Column direction: frequency. Solid magenta lines denote the auditory thresholds at different frequencies. Dashed red rectangles highlight columns of responses to different frequencies at the same SPL level (tonotopy). Het heterozygous, KO knockout. Colormap: parula (MATLAB). Top image array: an example from a P14.0 α9/α10 double heterozygous control. Bottom image array: an example from its P14.0 α9 knockout α10 heterozygous littermate. Schematics in the middle: major axes and tonotopic reversals in the two hemispheres of the IC. Box plots in Fig. 5: hinges: 25 percentile (top), 75 percentile (bottom). Box whiskers (bars): Max value (top), Min value (bottom). Source data and exact $p$ values are provided as a Source Data file.

presynaptic crossing fibers and postsynaptic auditory neurons, thus undermining bilateral circuit maturation. Moreover, the MOC-mediated bilateral coupling is itself tonotopic, consistent with the known efferent innervation pattern[19]. Therefore, the MOC-mediated cochlea coupling can serve as a neural substrate that mimics the bilateral features of real-world stimuli and promotes Hebbian plasticity across two sides of the auditory system. Indeed, Clause et al.[24] showed that MNTB-LSO (medial nucleus of the trapezoid body to lateral superior

olive) connectivity, where activity is integrated from both cochlea, is impaired in α9 nAChR knockouts, but not in the CN-MNTB calyces, where ipsilateral activity dominates. Behaviorally, these bilateral circuits are the neural basis for critical auditory functions such as sound localization[37]. α9 nAChR knockout animals also displayed deficits in frequency processing and sound localization at hearing onset[31]. Note that previous studies attribute these phenotypes solely to altered temporal patterns of spontaneous activity. We suggest that,

though precise temporal patterns may play an important role at the synaptic level, system-wide bilateral coupling informs auditory circuit refinement on a macroscopic scale. Together, we propose that spatiotemporal patterns of spontaneous activity can instruct auditory system maturation by recapitulating natural features of external inputs, which are nearly always bilaterally correlated. Interestingly, synchronous bilateral retinal spontaneous activity ("retinal waves") was also observed in the binocular zone of the visual midbrain (superior colliculi), possibly mediated by retinopetal or retino-retinal circuits[38–40]. From an evolutionary perspective, mammalian neural systems might precisely regulate spontaneous activity patterns to prime sensory circuits for future external inputs.

The second implication of these results stems from the developmental timeline for bilateral coupling, which might reflect changes of efferent modulation in vivo. Specifically, we observed that bilateral correlations peaked at the end of the first week after birth and then decreased through the second week after birth (Fig. 1k). P10–P11 has been described as a "transitional stage", during which MOC-OHC synapses form and MOC-IHC synapses start to decompose[14,17] and IHCs' responsiveness to acetylcholine starts to decrease[15]. At this stage, MOC mediated IPSCs are found in both IHCs and OHCs before withdrawing from IHCs later in the second week[29,41,42]. This general developmental trend is similar to the time course of bilateral coupling we observed (Fig. 1k). The chemogenetic rescue of bilateral coupling at P11 (Fig. 4j–l) suggests the persistence of MOC-IHC synapses during the "transitional stage" in vivo. Although certain synaptic properties, such as the size and replenishment rate of the readily releasable pool of synaptic vesicles, keep increasing from birth until ~P11, acetylcholine release from MOC fibers is negatively regulated by BK potassium channels after P9 and differentially regulated by different types of voltage-gated calcium channels before and after P9[29]. This suggests that MOC-IHC synapses at this late stage (~P11), though still functional, might normally be less effective than at earlier ages in modulating spontaneous activity in vivo.

Our results argue for a prominent developmental role of MOC-mediated efferent modulation of bilateral coupling, but they do not rule out the existence of other mechanisms that might also contribute to this phenomenon. In fact, our analysis showed residual bilateral correlations at least in the medial IC (future low-frequency region) and, in some cases, more lateral regions at P11–P12 (Fig.1k), in α9/α10 KOs (Fig. 2g), after acute pharmacology (Fig. 3f, Supplementary Fig. 4e), and after chemogenetic silencing (Fig. 4f; residual correlations summarized in Supplementary Table 1). Thus, efferent modulation of the IHC by the MOC produces robust bilateral correlations in the IC, but additional mechanisms may contribute to this phenomenon depending upon age and location.

Finally, MOC modulation might differ by tonotopic location. We utilized seed-based correlation maps to visualize bilateral correlation patterns, which also allowed us to differentiate correlation strengths at different future tonotopic locations. Although bilateral correlations showed similar temporal trends over-development (Fig. 1k), we found that future low-frequency regions were generally more bilaterally coupled than future high-frequency regions starting at P6–P7 (Supplementary Fig. 2f). This implies that MOC modulation might be stronger at the apical turn (low-frequency) than the basal turn (high-frequency region) in the cochleae. Indeed, in vitro electrophysiology data suggests that spontaneous firing of IHCs at the apical turn is more strongly modulated, presumably by the efferent system, than at the basal turn[12].

While we did observe a mild but significant increase in peak amplitude and event frequency in α9/α10 KO mice relative to

Hets (Fig. 2), we did not observe these differences after manipulating efferent modulation acutely (Figs. 4 and 5), despite the inhibition of IHCs by MOC efferents. Several factors might mask these effects. First, the magnitude of efferent modulation on IHCs is more pronounced when measuring correlations than the effect on amplitude, frequency or duration of events. Thus, the incomplete effects of DREADDs may be easier to observe on correlations than the other measures. Second, previous single-unit recording results suggest unchanged "overall firing rates" and "bursts/min" in the MNTB neurons of KO mice[24]. It is also possible that homeostatic mechanisms balance out these effects in the inferior colliculus relative to the MNTB. Third, KO mice display shorter burst firing durations, but they also have a higher firing rate during a burst[24]. These two changes may offset each other when imaging the IC at mesoscopic scales. $Ca^{2+}$ signals (based on GCaMP6s) also have rather slow dynamics, with the half decay time of GCaMP6s ranging from one to several seconds (in response to different numbers of action potentials[43]). The intense bursting spike pattern recorded in the α9 KO mice would prolong calcium signal dynamics compared to WT mice, potentially offsetting shortened burst durations. Finally, electrophysiology results acquired at the single-cell level in vitro are quite different than the global properties of hundreds of mesoscopic spontaneous bands measured in vivo. Even a single spontaneous band reflects the collective activity of thousands of neurons, bringing in further heterogeneity. Therefore, the dynamics of mesoscopic spontaneous bands might not be sensitive enough to reflect changes at the single neuron level.

In mature animals, MOC efferent fibers inhibit OHCs and suppress cochlear amplification[44]. Here we reported elevated auditory thresholds in the α9/α10 knockout at the hearing onset across the frequency spectrum (Fig. 5). Although MOC-OHC synapses are already functional at this age[42], the deficit is probably not caused by loss of acute MOC inhibition on the cochlear amplifier. In fact, eliminating the efferent modulation does not change auditory sensitivity as knocking out the α9 nAChR or the coupled SK2 channel does not alter ABR thresholds in mature animals[22,44], while strengthening efferent inhibitions leads to threshold elevation[45]. On the other hand, animals should have gained very limited auditory experience at this early age, as auditory thresholds are extremely high before P14.0[30] (see also Supplementary Fig. 6d). Taken together, our results suggest that threshold elevation in the α9 nAChR knockouts at hearing onset is primarily a developmental ramification. Note that this observation is based on central responses from the IC using wide-field calcium imaging, which is difficult to detect with the ABR method (Fig. 5b, c). Thresholds determined with the only visible ABR waves I–II are largely normal in the α9 nAChR knockouts compared to controls (Fig. 5a, b), consistent with data from adults[22]. Our results suggest that central circuits downstream of the cochlea nucleus, rather than the peripheral machinery, are undermined in the knockout, which is in line with previous reports of impaired tonotopic refinement and synaptic function in the MNTB with either the α9 nAChR knockout[24] or the enhanced α9 nAChR knock-in[32]. However, a more in-depth investigation of the degree to which the sensory periphery is intact in KO mice is needed for a more definitive conclusion, and it is unknown whether hearing thresholds will recover with auditory experience after hearing onset. Nevertheless, accumulated evidence stresses the important role of precise spontaneous activity patterns in facilitating auditory circuit maturation and the development of an auditory function.

We observed similar ABR thresholds in control animals as previously reported[30], but lower thresholds at most frequencies measured with calcium imaging (Supplementary Fig. 6e). There

are significant differences between the experimental set-ups of the two modalities (ABR and wide-field imaging). Wide-field imaging captures slow calcium activity on the scale of 50–100 ms, limited by the dynamics of GCaMP sensors, while the ABR captures fast electrophysiological dynamics on the scale of <1 ms. Accordingly, acoustic stimuli to induce calcium responses were usually much longer than those used in the ABR[30,46–48]. In our experiments, animals were fully awake during wide-field imaging of acoustic stimuli, sinusoidal-amplitude modulated tones of 500 ms duration were presented, and average calcium responses were acquired over 6-10 repeated sessions; for the ABR, animals were anesthetized, pure tone pips of 3 ms duration were presented, and average ABR waves were acquired over 400 repeats. Moreover, although the acoustic systems for the ABR and the calcium imaging were calibrated to have similar response curves, the exact apparatuses and background noise levels were still different (see "Methods"). Therefore, auditory thresholds measured with the ABR cannot directly translate to those measured with wide-field calcium imaging. However, our results did demonstrate that (1) central auditory responses in the IC can be detected by calcium imaging while only ABR waves I–II were consistently visible at P14.0 (Fig. 5), and (2) wide-field imaging could potentially capture lower auditory thresholds at hearing onset based on responses in higher auditory nuclei than the levels previously considered in the field. In fact, decades of psychoacoustic experiments on humans and animals indicate that behavioral thresholds can be ~20–30 dB lower than those measured with the ABR[49–52]. Therefore, it is possible that central auditory circuits can pick up signals from just a few peripheral neurons through a cascade of amplification and generate activity detectable with calcium imaging while remaining elusive to the ABR. Our results suggest that wide-field calcium imaging is a complementary technique for measuring auditory responses, especially in higher-order auditory nuclei.

Unilateral MOC neurons receive inputs from and provide feedback to both cochleae[26,44]. Functionally, the presentation of ipsilateral sounds can induce ipsilateral and contralateral MOC reflexes in mature animals[36,53]. We showed that single-side injections of the retrograde virus at P0–P1 can target olivocochlear neurons on both sides, confirming the existence of bilateral efferent circuits in neonates (Supplementary Fig. 5c). The distribution of labeled neurons resembled what had been described as the "DPO (dorsal preolivary)" and/or "shell" neurons[34,54]. An important question is: What drives these MOC neurons during the prehearing period? We propose two models here (Fig. 6): First, MOC neurons may be driven by the cochleae. This is the case in mature animals, where dominant inputs to the MOC neurons ultimately come from hair cells[26,36,55–57]. Second, there may be other common drives. The MOC neurons are innervated by different types of dendrites, suggesting heterogeneous inputs[58]. A recent study shows that MOC neurons receive inhibition directly from the MNTB[59]. Therefore, it is possible that other auditory nuclei may coordinate MOC neurons in both hemispheres. Note that these two models are not mutually exclusive and could work in synergy.

Prehearing auditory spontaneous activity exhibits different firing properties at different ages in vitro[11,60,61]. In comparison to the step-wise development of the visual system[40,62–64], it is intriguing to consider whether auditory spontaneous activity develops in a similar stepwise or continuous/smooth fashion. In the mouse cochlea, supporting cells display increasing firing amplitude, intracellular calcium transient level, and mechanical crenation magnitude from birth until ~P10. IHCs, driven by adjacent supporting cells, show a similar monotonic increase of spontaneous activity levels over the same period. Spontaneous firing in both cell types drops significantly at P13[11]. Comparable trends are observed in mouse cochlear nucleus (CN) and cat auditory nerves[60,61]. Here, we report similar developmental changes regarding spontaneous bands in the IC, as in vivo activity levels increased from P0 to P12 and then decreased at P13 (Fig. 1d, e). On the other hand, IHCs stay responsive to ATP during the prehearing period, suggesting that auditory spontaneous activity might be initiated by the same purinergic machinery before hearing onset[11]. We also showed that spontaneous events in the IC generally displayed band-shape intensity profiles with relatively smooth spatiotemporal variations, rather than crisp transitions across ages (Fig. 1 and Supplementary Movie S1). Taken together, in vivo and in vitro data suggests a continuous progression of the spatiotemporal features of auditory spontaneous activity during the prehearing period, in contrast to the mechanistically distinct stages defined in the visual system.

Tonotopy is the characteristic topographic organization of the auditory system. Recent imaging studies directly visualize tonotopic modules in auditory centers[23,47,48]. In the inferior colliculus, DCIC and LCIC display reverse tonotopic gradients at the dorsal surface,

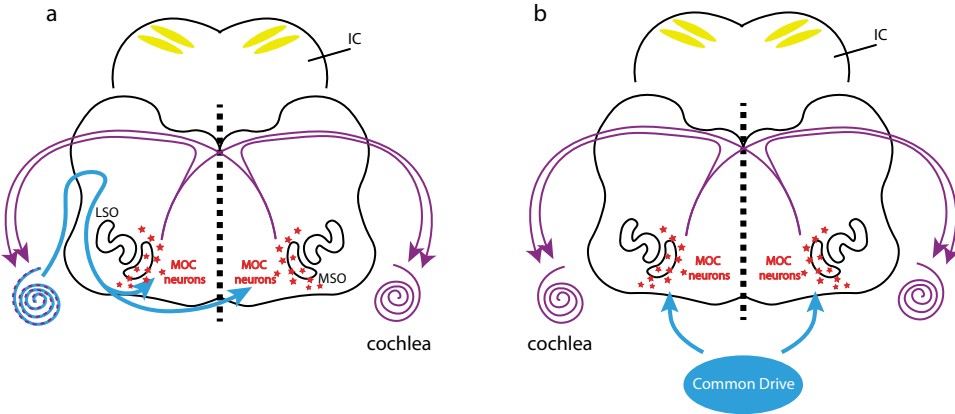

**Fig. 6 Models of MOC-mediated bilateral coupling of spontaneous activity. a** Model of the medial-olivocochlear (MOC) efferent circuits mediating bilateral coupling of spontaneous activity. MOC neurons are driven by and provide bilateral feedback to both cochleae, thus coupling the two sides of the auditory system. Schematics are at the level of the brainstem section where MOC neurons reside. LSO lateral superior olive, MSO medial superior olive. Purple curved arrows: cochleae receiving bilateral MOC modulation. IC is shown in the background with bilateral calcium bands presented. Driving inputs from cochleae to the MOC labeled as blue curved arrows. **b** Similar model as in (**a**). In this model, bilateral coupling derives from an external, common source (labeled in blue) to MOC neurons that is bilateral.

with low-frequency zones in the medial and high-frequency zones in the lateral parts[8,23]. We resolved this mirror-symmetrical structure from prehearing spontaneous activity with dimensionality reduction techniques (Supplementary Fig. 1k), suggesting the existence of dual-functional domains before hearing onset.

Intrinsically generated spontaneous activity is thought to play an essential role in the development of neural circuits, not just the presence of activity, but the very pattern of activity is instructive for setting up proper wiring schemes[2,65,66]. Despite this essential role, it remains largely unexplained what information these activity patterns carry, especially at macroscopic scales. Our results demonstrate that the medial olivocochlear efferent system provides strong bilateral coordination of spontaneous activity in the developing auditory system before hearing onset that is critical to the development of a normal auditory function. External auditory inputs are bilaterally correlated by nature. We propose a new role of efferent modulation during the highly plastic prehearing period: It preserves fundamental features of bilaterally correlated sound, instead of confusing developing circuits with two independent streams of information from each cochlea. We believe the bilateral coupling mechanism might serve as an evolutionary memory to prepare the immature auditory system for bilaterally coordinated sound immediately upon hearing onset.

## Methods

**Transgenic models.** SNAP25-G6s animals: B6.Cg-Snap25tm3.1Hze/J (JAX#025111) animals that have pan-neuronal GCaMP6s expression were used widely in this study. α9/α10;SNAP25-G6s animals: α9/α10 nAChR double-knockout (α9$^{−/−}$ α10$^{−/−}$) animals on a C57BL/6 background were obtained from Dr. Barbara Morley[22] (Chrna9tm1Bjmy MGI:5787807; Chrna10tm1Bjmy MGI:5787808). α9$^{−/−}$ α10$^{−/−}$ line was crossed to SNAP25-G6s line (JAX #025111) to generate double-heterozygous animals that express GCaMP6s (α9$^{+/−}$ α10$^{+/−}$; SNAP25-G6s$^{+/null}$). These animals were backcrossed to an α9$^{−/−}$ α10$^{−/−}$ line to generate four different genotypes of offspring (double-heterozygous: α9$^{+/−}$ α10$^{+/−}$, single-knockout: α9$^{+/−}$ α10$^{+/−}$, α9$^{−/−}$ α10$^{+/−}$, double-knockout: α9$^{−/−}$ α10$^{−/−}$) with or without GCaMP6s expression (SNAP25-G6s$^{+/null}$ or SNAP25-G6s$^{null/null}$). ChAT-Cre;SNAP25-G6s animals: Homozygous ChAT-Cre$^{+/+}$ animals (JAX #018957) were crossed to SNAP25-G6s animal (JAX #025111) to generate ChAT-Cre heterozygous offspring that express GCaMP6s (ChAT-Cre$^{+/−}$; SNAP25-G6s$^{+/null}$).

**Animals usage.** Animals of both sexes were used in this study. Animal care and use followed the Yale Institutional Animal Care and Use Committee (IACUC), the US Department of Health and the Human Services, and institution guidelines. In Fig. 1, SNAP25-G6s animals between P0 and P13 were used for spatiotemporal and correlation analysis at different ages. In Fig. 2 and related Supplementary Fig. 3, GCaMP6s-positive α9/α10;SNAP25-G6s animals (see Transgenic Models) were used for spatiotemporal and correlation analysis at P6–P7 or at P3–4. In Fig. 3 and related Supplementary Fig. 2, SNAP25-G6s animals between P5 and 7 were used for in vivo pharmacological experiments. In Fig. 4, GCaMP6s-positive ChAT-Cre; SNAP25-G6s animals (see Transgenic Models) were used for chemogenetic experiments. In Fig. 5 and the related Supplementary Fig. 4, the control group consists of wide-type SNAP25-G6s animals and α9/α10 double heterozygous animals. The α9/α10 knockouts consist of single and double knockout of α9 and/or α10 subunits. GCaMP6s positive animals were used for auditory thresholds measurement with wide-field imaging. Their GCaMP6s negative littermates were used for auditory brainstem response.

### Surgery for in vivo imaging

*Installation of cranial windows.* Mice were installed with cranial windows using procedures similar to ref. [40]: In brief, mice were anesthetized with isoflurane (2.5%) in oxygen and dorsal midbrain including superior and IC was exposed by a craniotomy. Noticeable modifications: (1) All animals were head-fixed on an articulating base stage (SL20, Thorlabs) with optic posts and angle post clamps that allowed rapid angular positioning. (2) To expose the entire dorsal surface of IC, the skull over IC and anterior-dorsal part of the cerebellum was removed. (3) For animals younger than P4, the initial isoflurane concentration for anesthesia was adjusted to 2% instead of 2.5%. (4) As spontaneous activity started to recover ~30 min after anesthesia[40], all animals were allowed at least 1 h to recover (with oxygen delivered) before imaging sessions. (5) Animals were nested with cotton gauze as previously described[63]. (6) For experiments involving bilateral A1 and IC (Figs. 2–3), the skull over A1 was rinsed with 1× phosphate-buffered saline (PBS) and cleaned with PVA eye spears. A minimal amount of cyanoacrylate glue was carefully applied to the cleaned skull to create a smooth surface for direct imaging through the skull.

*In vivo pharmacology via the round window.* After the acquisition of non-manipulated data, animals have anesthetized with 2.5% isoflurane again. Articulating stage was tilted ~45° and angle post clamps were adjusted to allow better angular positions to perform postauricular incisions. Procedures to expose round window niches are similar to ref. [8]. To facilitate robust pharmacological delivery, round window membrane was punctured and removed using fine forceps. Fluid efflux was drained with sterile filter paper. Gel form soaked with 1 μL of different pharmacological compounds (apamin: 200 μM; alpha-bungarotoxin: 1 mM; or 1× PBS. Apamin: Tocris, Cat#1652; Alpha-Bungarotoxin: Abcam, Cat#ab120542) was shallowly inserted into the cochlea through the round window. The opening was quickly sealed with cyanoacrylate glue using a plastic glue stick. The surgery was performed on both sides of the cochlea in ~20 min (isoflurane was changed to 1.5% during the surgery and stayed off during recovery). Animals were allowed at least 1 h to recover (with oxygen delivered) before imaging sessions.

**In vivo calcium imaging.** Schematics of wide-field imaging apparatus is shown in Fig. 1a. The microscope was placed in a soundproof booth as described in Acoustic Stimulation For In vivo Calcium Imaging, this paper. Procedures for imaging were similar to ref. [8]. In brief: an sCMOS camera (pco.edge, PCO) coupled to a Zeiss AxioZoom V16 microscope with a 1× macro objective was used for all calcium imaging. Noticeable modifications: (1) Pups were wrapped with cotton gauze during data acquisition as described in ref. [63] instead of being placed in a swaddling 15 mL conical centrifuge tube. (2) The size of the field of view varied across conditions to maximize the visibility of ROI (IC only or simultaneous imaging over auditory cortex and IC). (3) Each recording session contained continuously acquired movies for more than 30 min.

### Chemogenetic experiments

*PSCC injection in neonates.* Neonatal ChAT-Cre;SNAP25-G6s animals (see Transgenic Models) were genotyped and GCaMP6s-positive pups were selected for PSCC injection at P0–P1. Procedures were similar to ref. [67] with modifications: (1) Pups were placed in crushed ice for ~90 s for initial anesthesia. (2) Athetized pups were placed on a reusable gel ice pack with crushed ice around the body. (3) 2 μL (1 μL each side) of floxed retrograde AAV was injected (AAVrg-hSyn-DIO-hM4D-mCherry: titer ≥ 8 × 10$^{12}$ vg/mL, #44362-AAVrg, Addgene; or AAVrg-hSyn-DIO-hM3D-mCherry: titer ≥ 7 × 10$^{12}$ vg/mL, #44361-AAVrg, Addgene) via micro-injector. All procedures were finished in 10 min.

*Clozapine N-oxide (CNO) injection.* CNO (Tocris, Cat# 4936) dose was 5 mg/kg for inhibitory-DREADD (hM4Di) and control experiments (Fig. 4a–f, Extended Fig. 5e–h) and 1 mg/kg for excitatory-DREADD (hM3Dq) experiments (Fig. 4g–l). After acquisition of non-manipulated data, a fixed volume of CNO (various concentrations to match body weights with required doses) was injected intraperitoneally (IP injection) to animals. As reported previously[68], an adequate amount of CNO can be detected in brain tissues 15 min after injection. Imaging sessions started ~15 min after IP injection and lasted for 30 min.

*Immunohistochemistry.* Mice were anesthetized via IP injection with a "rodent combination cocktail" (Ketamine 37.5 mg/ml, Xylazine 1.9 mg/ml, and Acepromazine 0.37 mg/ml) at 1 ml/kg and perfused transcardially with 1× PBS followed with 4% PFA. Brains were removed and fixed overnight in 4% PFA. After embedding brains in 2.5% agarose (made in 1× PBS), coronal sections (50um) were collected using a vibratome (Leica VT1000 S). Brain slices were permeabilized in 0.7% Triton X-100 in 1× PBS for 30 min before blocking. Permeabilized slices were rinsed in PBST (1× PBS with 0.01% Triton X-100) three times (10 min each time) and incubated with blocking solution (10% donkey serum, 1% bovine serum albumin, 0.5% Triton X-100 in 1× PBS) for 24 h at 4 °C. Brain slices were rinsed three times in PBST and then incubated with primary antibody (1:400; goat anti-choline acetyltransferase (ChAT), catalog# AB144P, Millipore) for ~48 h at 4 °C. After primary incubation, the slices were rinsed three times in PBST and then incubated with secondary antibody (1:500 Alexa 647-conjugated Donkey Anti-Goat IgG, Cat# 705-605-147, Jackson ImmunoResearch) at room temperature for ~3 h. After secondary incubation, the slices were rinsed 3 times in PBST and mounted on glass slides with an antifade mounting medium with DAPI (VECTASHIELD). Images were acquired using a Zeiss Axio Imager Z2 equipped with a CCD camera (AxioCam HRC, Carl Zeiss). Merged images were constructed using Zeiss ZEN Blue software from four different channels with pseudocolors (Supplementary Fig. 5: Blue: DAPI; Magenta: GCaMP; Red: mCherry; Green: Alexa 647).

**Acoustic stimulation for in vivo calcium imaging.** A wide-field microscope was enclosed in a double-pane booth made of soundproof plywood and acoustic foam (See Fig. 5d). Background noise was <30 dB SPL inside the booth (the major source was the CMOS camera fan at the top of the wide-field microscope, measured with a BK Precision 735 sound level meter). An electrostatic speaker (ES1, Tucker-Davis Technologies) was placed 5 cm from lambda and parallel to the sagittal suture. Acoustic stimuli were generated with customized open-source software in MATLAB ("Baphy", Neural Systems Laboratory, University of Maryland College Park). A digital-to-analog converter (National Instruments BNC-2110) transformed MATLAB outputs to voltage signals to the speaker driver (ED1,

Tucker-Davis Technologies). To optimize calcium responses in wide-field imaging setting[47], sinusoidal amplitude-modulated tones (SAM, modulation depth = 1; modulation frequency = 10 Hz) were presented free field at frequencies between 4 and 32 kHz (at half-octave step) and sound intensities from 96 dB SPL to 8 dB SPL (in 8 dB decrements). This resulted in 84 different frequency-SPL combinations. Each stimulation trial was 5-s long consisting of 1-s prestimulation idle time, 0.5-s SAM acoustic stimulation, and 3.5-s post-stimulation idle time. Trials were spaced by random intervals (between 0 and 5 s). Camera frames were triggered by and time-locked with acoustic stimuli via a neurophysiological stimulator (Master 8, A. M.P.I. Israel) so that each simulation trial rendered exactly 50 camera frames (movies were acquired at 10 Hz). One stimulation session contained all 84 frequency-SPL combinations in random order and produced a 4200-frame (420-s) movie. Digital signals from MATLAB, analog signals (voltage inputs) to the speaker driver, and camera feedback were recorded in a multi-channel data acquisition platform (SPIKE2, CED) at a 125 kHz sampling rate for quality control and data analysis. An ultrasound microphone (type: USM EK-FG USG. Connected to an UltraSoundGate 116Hb recording interface, Avisoft Bioacoustics, version 4.2) was placed 5 cm away from the ES1 speaker to measure SPL for response curve calibration (Knowles FG, Avisoft Bioacoustics). Sound levels (pure tones) produced by the ES1 speaker were calibrated to be consistent with the outputs of the FF1 speaker described in the ABR section (±3 dB error). Each animal was presented with 6–10 stimulation sessions in a total of 1–2 h.

To pinpoint birth time accurately, cages were checked twice a day (~12 h apart). Acoustic stimulation experiments were conducted at P14.0 (within 12 h interval from $13 \times 24$ h to $13 \times 24 + 12$ h after birth), P13 (within 24 h interval from $12 \times 24$ to $13 \times 24$ h after birth), or P12 (within 24 h interval from $11 \times 24$ to $12 \times 24$ h after birth).

**Auditory brainstem response (ABR)**. ABR experiments in this study were conducted using the same apparatus (TDT3 system with an FF1 speaker, Tucker-Davis Technologies) and procedures described in ref. [69,70]. ABR waves were analyzed by the BioSigRP software (Tucker-Davis Technologies, version 4.4.10). Auditory thresholds were determined based on waves I and II (see also Fig. 5a). Genotype information was blinded to the tester. In brief, stimuli were presented free field 10 cm from the vertex. Frequencies ranged between 2 and 32 kHz (at half-octave step) and sound intensities decreased from 105 dB SPL (at 5 dB step). Response waveforms were acquired by averaging over 400 repeats for each condition. Tone bursts were 3 ms long (1 ms raised cosine on/off ramps and 1 ms plateau) and were delivered at a rate of approximately 20 per second. A bandpass filter (50–3000 Hz) was applied during recording. ABR threshold was defined as the lowest intensity (to the nearest 5 dB) capable of evoking a reproducible, visually detectable response.

**Data analysis**

*Image preprocessing*. Raw TIFF movies were preprocessed with an object-oriented pipeline[71] (https://github.com/CrairLab/Yixiang_OOP_pipeline). Parallel computing was realized in MATLAB on Yale's high-performance clusters (Yale Center for Research Computing). The pipeline included the following steps (see Supplementary Fig. 1a): (1) Photobleaching correction using a single-term exponential fit on averaged fluorescent intensity (frame-wise). (2) Motion correction/subpixel rigid registration[72]. (3) Gaussian smoothing (sigma/filter size = 1). (4) ROI mask application (manually defined in ImageJ). (5) Downsampling (average-pooling with a $2 \times 2$ filter). (6) Denoising based on singular-vector decomposition. (7) Intensity normalization ($\Delta F/F_0$, where $F_0$ was defined as the 5th percentile value for each pixel). Key parameters were kept consistent across conditions.

*Seed-based correlation analysis*. A number of reference seeds were pre-defined before parallel processing. Movies with a down-sampled field of view were densely covered by 1000 evenly spaced seeds. Regular or partial Pearson-correlation matrices were generated with seeds inside ROI masks using built-in MATLAB functions "corr" and "partialcorr" respectively (see Supplementary Fig. 2). For partial-correlation matrices, correlations between reference seeds and other pixels in the ROI were controlled for the averaged fluorescence trace over all pixels outside the ROI (to regress out the influence of non-specific whole-brain fluctuations). All seed-based correlation analysis in this study is based on partial Pearson-correlation except for the demonstration in Supplementary Fig. 2a. All seed-based correlation maps were manifested with jet (256) colormap and [−0.2, 1] color limit except for the demonstration in Supplementary Fig. 2a (with [−1, 1] color limit to better illustrate near-zero correlations). Frames identified as containing motions (when amplitudes of subpixel rigid registration were larger than 0.5 pixels) were excluded using motion correction information generated in preprocessing procedures.

GraphPad Prism (RRID:SCR_002798) and Adobe Illustrator (RRID: SCR_010279) were used to generate/organize figures.

**GUI for wide-field auditory data analysis**. A novel graphic–user-interface ("Manuvent" GUI[73], see Supplementary Fig. 1c, https://github.com/CrairLab/Manuvent) was designed for interactive data analysis based on pre-processed movies. This multi-purpose platform allowed the following analysis.

*Line scan and peak detection*. Rectangular ROIs were interactively drawn in the GUI. Cropped movies were averaged over the direction parallel to the major axis of spontaneous bands to reduce two-dimensional frames to one-dimensional linear representations across the tonotopic axis (see Supplementary Fig. 1b[8]). This process reduced a three-dimensional movie (two-dimensional frames × one-dimensional time) to a two-dimensional line-scan map (one-dimensional line representations × one-dimensional time). 3D spatiotemporal profiles of individual spontaneous bands were represented as two-dimensional $\Delta F/F_0$ bumps in the line-scan maps (Supplementary Fig. 1b). Peak detection of spontaneous events (finding the local maximum of $\Delta F/F_0$ bumps) was accomplished in an auxiliary GUI ("LineMapScan", see Supplementary Fig. 1b). In brief, local maxima were detected by scanning along with the spatial and temporal directions separately with MATLAB built-in function "findpeaks" and coinciding the partial results to identify spatiotemporal peaks (when $\Delta F/F_0$ values reach maxima in two directions simultaneously). Line-scan maps were smoothed with a $5 \times 5$ spatiotemporal filter (much smaller than minimum spatial and temporal half-width of spontaneous events across ages) before peak detection. 5% $\Delta F/F_0$ threshold was consistently used across conditions. Events corrupted by motions (epochs where amplitudes of subpixel rigid registration were larger than 0.5 pixels) were automatically excluded using motion correction information generated in preprocessing procedures.

*Quantification of spatiotemporal properties*. The following statistics were automatically quantified within the GUI: (1) Event frequency (number of individual spontaneous bands per minute). (2) Event duration (mean duration of spontaneous bands, defined as the half-width of temporal peaks). (3) Inter-peak-interval (IPI, average interval between two neighboring peaks). Note that when quantifying IPIs, multiple peaks from the same frames were counted as a single event to exclude zero-length intervals. (4) Normalized bandwidth (defined as the half-width of spatial peaks normalized by the width of the IC). (5) Peak amplitude (average fluorescent intensities at peaks).

*Global bilateral correlation analysis*. The left and right hemispheres of IC were manually delineated in the GUI. Mean hemispherical fluorescent traces (averaged over pixels in each hemisphere) were plotted and compared in the GUI (see Supplementary Fig. 1c, g). Partial correlation between the two average traces was defined as global bilateral correlation and was computed with built-in MATLAB function "partialcorr" controlling the mean fluorescence trace over all pixels outside the IC. This analysis was controlled for motions as described before (see Seed-based correlation analysis, this paper).

*Manual event-labeling*. Pre-processed movies were manually assessed in the Manuvent GUI. A tester was asked to label the first and last frames of individual events by clicking at the best-estimated centers of spontaneous bands up to the human-eye threshold. $\Delta F/F_0$ matrices were converted to gray-scale movies with built-in MATLAB function "mat2gray" (fixed grayscale limit of [0, 0.3] across conditions to avoid artificial visual amplification induced by automatic scaling), and displayed in the GUI. The tester was trained on a separate set of five movies that were not included for final analysis until the resulting statistics (number of bands per minute, and mean event duration) stabilized after labeling the same set of Movies 6 times in random order. The GUI allowed users to play movies forward/ backward, frame-by-frame, or check/edit labeled events interactively. Movies acquired in different experimental conditions were shuffled and the tester was blinded to any prior information (age, genotype, weight, sex). Summary statistics were based on manually labeled events in both hemispheres of IC.

**GUI for seed-based correlation analysis**. A novel GUI ("PickSeedMap" GUI[74], see Supplementary Fig. 2b, https://github.com/CrairLab/PickSeedMap) was designed for correlation map visualization, representative maps selection, and analyzing regional properties of correlation patterns. Seed-based correlation matrices were generated with the pipeline mentioned above.

*Selection of representative seeds*. Three representative seeds corresponding to low-/mid-/high-frequency regions were chosen based on typical correlation patterns seen in P3–P7 control animals. Summarized criteria were used across conditions: (1) The first representative seed (corresponding to the low-frequency region) was selected when only a single high-correlation band was visible in each hemisphere (see Fig. 1h left panel, Supplementary Fig. 2c, and Supplementary Movie S2). (2) The next representative seeds were chosen 3–4 seeds down relative to the previous ones where a pair of high-correlation bands were visible in each hemisphere (see Fig. 1h middle and right panels, Supplementary Fig. 2c and Supplementary Movie S2). (3) Representative seeds were all located in the medial region of the left IC. For experiments with large field of view (simultaneous A1/IC imaging), a single representative seed in the mid-/ low-frequency region of the left IC was selected for quantifying IC-contralateral-IC, IC-ipsilateral-A1, IC-contralateral-A1 correlations (Fig. 2h, i). Another representative seed in the left A1 was selected for quantifying A1-contralateral-A1 correlation (Fig. 2h, i).

*Determine maximum correlation in a region*. To determine the maximum correlation value in a region of interest (usually the contralateral hemisphere) with respect to a given seed, a large ROI was defined symmetrically (Fig. 1h, i and Supplementary Fig. 2b). The pixel within the ROI whose immediate neighborhood hosted the highest mean correlation (using a $5 \times 5$ averaging filters) was identified,

and this means correlation value was defined as the maximum correlation in the ROI w.r.t the seed.

*Compute regional properties of correlation patterns.* The Nearby region of the first representative seed (corresponding to the low-frequency region) was converted to a connected component using a threshold of 0.95 (selecting pixels that had >0.95 correlation w.r.t. the seed). The connected component was approximated as an ellipse (see Supplementary Fig. 2c: left panel), and its regional properties were extracted using a built-in function "regionprops" in MATLAB. Eccentricity of this region was quantified as the aspect ratio of the approximated ellipse (a/b). The proportion was calculated as the area of the region normalized by the area of the hemisphere.

**Determine auditory thresholds from calcium data**. Animals were presented with 6–10 sessions of 84 different frequency–SPL combinations of SAM tones (see Acoustic Stimulation For In vivo Calcium Imaging, this paper). All movies were pre-processed with the same object-oriented pipeline mentioned above (see Image Processing, this paper) leaving out the normalization ($\Delta F/F_0$) step. Trials in different sessions but presented with the same frequency–SPL tone were sorted to the same group. Frame-wise averages were computed across all trials within the group. The preprocessing pipeline outputted 84 mean-response matrices/movies (each contained 50 averaged frames). Normalization ($\Delta F/F_0$) was conducted after getting the mean-response matrix and $F_0$ was defined as the average fluorescent value of the first 10 frames (before acoustic stimuli were presented) for each pixel. Using the "Manuvent" GUI (see GUI For Wide-field Auditory Data Analysis, this paper), the pixel that had maximum $\Delta F/F_0$ values between frame #11–15 (during which the acoustic stimuli were presented) was automatically detected (see Supplementary Fig. 6b, left panel: the red dot denoted the max-responding pixel). To obtain a robust response curve based on population activity, a seed-based correlation map was generated with respect to the max-responding pixel using the "PickSeedMap" GUI (see GUI For Seed-based Correlation Analysis, this paper). The populational activity was pooled from pixels within the high-correlation (>0.997) region (see Supplementary Fig. 6b, right panel: the region in white). The mean response curve was defined as the average fluorescent trace across all pixels in the high-correlation region (see Supplementary Fig. 6b). Animals were considered responding to a tone at a certain SPL level if the curve crossed the 2% $\Delta F/F_0$ threshold during frame #11–20 (see Supplementary Fig. 6c). The lowest SPL at which animals responded to the tone was defined as the auditory threshold of the tone. The same criteria were applied across conditions in this study to determine animals' auditory thresholds. *Generate average response images*: Example response images shown in Fig. 5c and Supplementary Fig. 6c were averaging over frames #11–15 from all sessions during which acoustic stimuli were presented.

**GUI for dimensionality reduction and unsupervised clustering**. A novel GUI (GUI_dimReduction[75], see Supplementary Fig. 1i, https://github.com/CrairLab/GUI_dimReduction) was designed for dimensionality reduction, data visualization, and unsupervised clustering of wide-field calcium imaging data based on two commonly adopted techniques: diffusion map[76] and t-SNE[77]. All dimensionality reduction analysis in this study was based on a diffusion map.

*Dimensionality reduction and data visualization.* Pixels in a field of view were projected to and visualized in three-dimensional space by conducting diffusion map analysis solely based on their activity (fluorescent traces). In brief, a Gaussian kernel was used to project data points (vectors representing the fluorescent activity of pixels) to a weighted graph. The analysis modeled data similarity by conducting random walks on the constructed Gaussian-weighted graph and computing eigenfunctions of the corresponding Markov matrix. The eigenfunctions were used to compute diffusion coordinates and distances, which quantified similarities of the original pixels in terms of their activity. For diffusion map analysis, sigma values that determined diffusion speeds on Markov networks were defined as two times the standard deviation of pairwise distances among all pixels (Euclidean distances between pixels in the activity space) in order to adapt with different data landscapes. An auxiliary GUI ("CorrespondMaps") allows users to correspond projected points in the low-dimensional space back to the original pixels interactively (using built-in brushing function in MATLAB, see Supplementary Fig. 1j).

**Reporting summary**. Further information on research design is available in the Nature Research Reporting Summary linked to this article.

## Data availability
Raw data are available from the corresponding author (michael.crair@yale.edu) upon reasonable request. Source data are provided with this paper.

## Code availability
All code used for analysis is available through Github. Specifically: (1) Object-oriented pre-processing pipeline for wide-field auditory data analysis[71] (https://doi.org/10.5281/zenodo.4516001; latest version: https://github.com/CrairLab/Yixiang_OOP_pipeline). (2) Graphic–user-interface for wide-field auditory data analysis[73] (https://doi.org/10.5281/zenodo.4515995; latest version: https://github.com/CrairLab/Manuvent). (3)

Graphic–user-interface for seed-based correlation analysis[74] (https://doi.org/10.5281/zenodo.4515999; latest version: https://github.com/CrairLab/PickSeedMap). (4) Graphic–user-interface for dimensionality reduction and unsupervised clustering[75] (https://doi.org/10.5281/zenodo.4515989; latest version: https://github.com/CrairLab/GUI_dimReduction).

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

## Acknowledgements

We would like to thank Dr. Dwight Bergles and the Bergles lab at Johns Hopkins University, all members of the Crair lab, the Santos-Sacchi lab, and the Navaratnam lab for their helpful comments on this project. We would like to thank Dr. Leonard Kaczmarek and Dr. Yalan Zhang for providing apamin for pilot pharmacological experiments. We would like to thank Dr. Winston Tan and Dr. Jun-ping Bai for providing guidance on conducting ABR experiments. We would like to thank Dr. Steven W. Zucker and for suggestions on applying diffusion map analysis. We would like to thank Dr. Rui Chang and Chuyue Yu for providing Alexa Fluor® 647 AffiniPure Donkey Anti-Goat IgG (H + L) antibody. We would like to thank Dr. Matthew McGinley for setting up the "Baphy" software. We would like to thank Dr. Jessica Cardin for helpful discussions regarding this study, and the family of William Ziegler III for their support. This work was supported by NIDCD R01DC008860.

## Author contributions

Conceptualization: Y.W., A.G., and M.C.C.; Methodology: Y.W., A.G., L.S., and M.C.C.; Software: Y.W.; Formal analysis: Y.W. and M.S.; Investigation: Y.W.; Resources: B.M., J.S., D.N., and M.C.C.; Writing–Original Draft: Y.W.; Writing–Review and Editing: Y.W., M.S., L.S., A.G., D.B., J.S., and M.C.C.; Visualization: Y.W.; Supervision: M.C.C; Project Administration: Y.W., Y.Z., and M.C.C.

## Competing interests

The authors declare no competing interests.
