## [Peer Review File · Nature Communications]

Reviewers' Comments:

Reviewer #1:

Remarks to the Author:

This study reports that the cholinergic medial olivocochlear system (MOC) plays a role in the bilateral coordination of cochlear-generated spontaneous activity before hearing onset. The authors report that absence of cholinergic transmission to hair cells in mice that lack $\alpha 9$ or $\alpha 10$ nicotinic AchRs in hair cells or by pharmacological block of nicotinic AchR receptors, decreases the correlation of activity in the two IC hemispheres whereas increasing MOC activity increases correlations. Furthermore, in KO mice, the response threshold in the IC to sound after hearing onset is increased. These results suggest that the MOC system contributes to the bilateral coordination of spontaneous activity patterns before hearing onset.

The question whether spontaneous prehearing activity is coordinated between both cochleae is a long-standing question and of importance for our framework how binaural auditory circuits become properly wired. The study is well executed but I have some questions and comments I would like to see addressed.

The authors state that the developmental time course of bilateral correlations described in Figure 1 follows the developmental time course of effectiveness or strength of cholinergic transmission in inner hair cells (Kearney et al., 2019). However, looking at the cited paper I could not find such a correlation as Kearney et al. found a steady increase in transmission of MOC synapses up to P9-11, whereas the age-dependent correlation of IC activity forms a U-shaped curve that peaks at P3-4 and declines thereafter. Some more explanations seem warranted.

Ext fig 1 shows a halfwidth duration of bands of around 1.5 to 2 seconds and Fig 1f shows 50-80 bands occur per minute. This seems impossible – how is a band halfwidth of 1.5 second possible if inter band interval is much less than a second? This is especially pronounced in the P11-12 group. Also, in Ext Fig 1: There are two mirror-imaged tonotopic axes in IC. Since activity originates in a tonotopically restricted area in the cochlea should the analysis not take this into account two peaks per even, one for each tonotopic axis?

Fig 2: Is the correlation drop in $\alpha 9/\alpha 9$ KOs that is shown for P6-7 mice also present at other ages and do these KOs show a similar time course in peak amplitudes and event frequency?

A previous paper by Babola et al (Neuron, 2018) also reported some correlated activity bands in both IC hemispheres, which these authors attributed to a weak ipsilateral projection from the cochlear nucleus to the IC and the presence of commissural connections between both ICs. This interpretation was partly based on the observation that there was a strong difference in the amplitudes of coordinated activity bands between both ICs. Most importantly, even after unilateral cochlear ablation or unilateral pharmacological block of cochlear AMPA receptors, coordinated bilateral bands were still present, which indicates that one cochlea alone can drive both IC hemispheres. While these results do not exclude the possibility that MOC-mediated bilateral coordination exists, it indicates that the MOC is a contributor rather than the main driver of bilateral coordination. This is not made very clear in the discussion, title and introduction, the latter of which states that the MOC system is necessary for enforcing bilateral coupling. The authors could also address this by cutting the commissural connections and measuring the resulting correlation patterns. □□□□□

MOC synapses inhibit inner hair cells. Would it not be expected that activation of MOC would decrease the peak amplitude of IC bands or decrease their frequency, which however, was not the case (Fig 4h,i).

Similarly, previous recordings in prehearing $\alpha 9$ KO mice showed a significant shortening of activity bursts in these mice (Clause et al., 2014). Since the length of these bursts is expected to influence the duration of IC bands one would expect a decrease of band duration in the $\alpha 9/\alpha 10$ KO mice. The authors

analyzed and duration in WT mice but not KO mice but it would be interesting and informative to know whether band duration is reduced in the KO and CNO treated DREADDs mice. The authors already have the data, so the analysis should be included in the paper.

It is not quite clear why alpha-Bungarotoxin was applied to both cochleae. Should unilateral blockage not be sufficient for decorrelation?

It is becoming increasingly clear that CNO can have potential off-target effects. Thus, it is necessary to present the effect or lack thereof of administering CNO in WT animals.

Fig 5d is not very informative and can be deleted to make space enlarge 5C.

Looking at the traces of sound evoke activity in IC, it seems that the KO mice had a less bilaterally correlated activity in the IC than the Het mice. Given that there was free-field stimulation this is a bit surprising, if the images shown are representative.

The ABR traces look atypical compared to the waveforms that are generally published. This is especially obvious for the trace from the P60 animal that in which waves ride on each other instead of having propounded negative peaks.

Line 137: Local connectivity. The authors speculate the existence of developmental changes in local connectivity. The authors may be referred to Sturm et al., 2014, who investigated exactly this issue.

Often, the labeling for the look-up tables and significance bars and so small that it is undecipherable. Also, even if a simple computer screen shot is show in the figure, axes should be labelled.

Line 163 – “a9/a10” Het should read “a9 KO/a10 het”?

Reviewer #2:

Remarks to the Author:

The paper by Wang et al. charts the profile of bilateral coupling in the developing inferior colliculus (IC). The authors demonstrate that correlated spontaneous activity across the two hemispheres peaks several days prior to hearing onset, and that coupling is dependent on cholinergic feedback to the cochlea. Disruption of this pathway is associated with a loss of auditory sensitivity at hearing onset. This is a very interesting study, and I particularly admire the range of complementary approaches that have been used to demonstrate how altering cholinergic feedback to cochlea influences across-hemisphere correlations at the level of IC - genetic knockouts, pharmacology and chemogenetics. There are some aspects that require further clarification:

Major -

There is always a concern that changes in correlation strength are purely the result of changes in the number and/or amplitude of events. Here, sub-significant changes (or barely significant changes, i.e. Fig2D) in the frequency and amplitude of spontaneous calcium events could be leading to significant changes in correlation. Could the authors present a quantitative refutation of this idea? For example, they could break recording sessions down into epochs and then (across all animals) measure the relationship between event frequency (in these epochs) with the correlation values (likewise with the event amplitude) and, hopefully, show no relationship exists.

Why does altering feedback, via activation of excitatory/inhibitory DREADDs, not alter event frequency or event amplitude? If activation of the MOC, via spontaneous cochlea activation, is enough to drive synchronised activation of both hemispheres of the IC then why is fundamentally altering general

cochlea activity using chemogenetics not sufficient to alter event amplitude or frequency at all?

The movies look as though there are delays between left and right (or sometimes that intensity increase once both are active). Did the authors look for temporal delays between hemispheres to understand to what extent cross-talk influenced spontaneous events? – This might be important as I think they suggest the coupling happens at the input level not the IC level?

133-135: “In addition, we noticed a tonotopic difference in bilateral correlations using SbBCs, with low frequency regions exhibiting stronger correlations than those of high frequency regions at P6-7 (Extended Data Fig. 2d), suggesting that coupling strengths can vary by tonotopic location.” – Could this also be due to a better field of view of these areas?

236-237: It would be useful to include an estimate of how much thresholds changed (even if just a mean value across frequency).

242-243: “and there was little difference in responses in a9/a10 knockout and control animals, except at the lowest frequency (Fig. 5a)” - Here the authors use a t-test with multiple comparisons correction. Why not an overall statistic which would be more sensitive to general differences? If they are all slightly different it suggests that overall they might be significantly different.

439-440: “that is critical to the development of normal auditory function” – the data show that hearing thresholds are elevated very soon after hearing onset (P14). However, it is not known whether hearing thresholds can subsequently recover. The text should be amended to reflect this uncertainty.

Minor corrections:

Figure 1 – panel c is mislabelled; panel g requires time calibration bar.

Figure 4 – panels a, g – ‘Imged’ > imaged

Figure 5 – panel e – legend typo

line 381 – “lowers” = lower

line 393 – “Frist,” = first

Response to Referees

Note: Reviewer remarks in **green**. Author replies in **black**. Adjusted manuscript lines in **blue** (please note that we are referring to line numbers in the document with “**track changes**”).

Reviewer #1:

This study reports that the cholinergic medial olivocochlear system (MOC) plays a role in the bilateral coordination of cochlear-generated spontaneous activity before hearing onset. The authors report that absence of cholinergic transmission to hair cells in mice that lack $\alpha 9$ or $\alpha 10$ nicotinic AchRs in hair cells or by pharmacological block of nicotinic AchR receptors, decreases the correlation of activity in the two IC hemispheres whereas increasing MOC activity increases correlations. Furthermore, in KO mice, the response threshold in the IC to sound after hearing onset is increased. These results suggest that the MOC system contributes to the bilateral coordination of spontaneous activity patterns before hearing onset.

The question whether spontaneous prehearing active is coordinated between both cochleae is a long-standing question and of importance for our framework how binaural auditory circuits become properly wired. The study is well executed but I have some questions and comments I would like to see addressed.

1. The authors state that the developmental time course of bilateral correlations described in Figure 1 follows the developmental time course of effectiveness or strength of cholinergic transmission in inner hair cells (Kearney et al., 2019). However, looking at the cited paper I could not find such a correlation as Kearney et al. found a steady increase in transmission of MOC synapses up to P9-11, whereas the age-dependent correlation of IC activity forms a U-shaped curve that peaks at P3-4 and declines thereafter. Some more explanations seem warranted.

Author replies: Thank you very much for the feedback! The main point we are trying to make is that the general developmental trend we observed is consistent with previously published data. We do not intend to imply a tight day-by-day correspondence, as the techniques we are employing (imaging of the IC *in vivo*) are very different than the published *in vitro* MOC-IHC synaptic recordings. We have simplified and modified the text in our manuscript to make this point clearer (top of page 11, around lines 331). The published data (e.g. Kearney et al., 2019 and see the review by Simmons, 2002) report that MOC fibers transiently innervate IHCs during the first week after birth, and then transition to largely innervate OHCs by the end of the second week after birth. During this rather brief window of transient innervation, the detailed functional properties of the MOC-IHC synapse change significantly, with changes in voltage dependent Ca^{2+} channels, K-channels and the probability of release. These synaptic changes occur on the background of refinement in the number/density of efferent nAChR-mediated connections onto IHCs (Katz et al., 2004). Overall, this is consistent with the peak in bilateral correlations we observed *in vivo* around P7, and its diminution through P12, though the direct day-by-day correspondence between the *in vitro* properties of MOC-IHC synapses and *in vivo* correlations in the IC are difficult to establish.

2. Ext fig 1 shows a halfwidth duration of bands of around 1.5 to 2 seconds and Fig 1f shows 50-80 bands occur per minute. This seem impossible – how is a band halfwidth of 1.5 second possible if inter band interval is much less than a second? This is especially pronounced in the P11-12 group. Also, in Ext Fig 1: There are two mirror-imaged tonotopic axes in IC. Since activity originates in a tonotopically restricted area in the cochlea should the analysis not take this into account two peaks per event, one for reach tonotopic axis?

We quantified individual bands at different locations along the tonotopic axis separately. At a specific moment in time (frame in a movie), multiple bands can appear simultaneously. In other words, these bands “overlap” temporally but not spatially (they are located at different sites on the tonotopic axis). These individual bands can be distinguished and are examined separately by the analysis algorithm.

As the reviewer notes, there is a mirror-symmetric organization of tonotopic bands in the IC when viewed from above. This can be confirmed after pre-processing and correlation analysis, which requires an appropriately defined ROI that includes both tonotopic axes to begin with. Before running this analysis, it is difficult (and arbitrary) to manually define an ROI that only includes one of these two axes and confine the analysis to this axis because: 1) the two tonotopic axes (in reverse directions) meet in the medial IC, but one cannot determine by eye exactly where the boundary is before running the processing pipeline. 2) Spontaneous bands in some animals are slightly curved, making it even harder to manually define the ROI to accurately follow the curve. 3) The field of view contains duplicate tonotopic representations for some frequencies, but not for others, and this varies from animal to animal and even across hemispheres in the same animal. Therefore, the current analysis method was essentially fully automated and includes the entire field of view (both tonotopic axes), which we believe introduces fewer artifacts that could potentially skew the analysis. The same automated analysis method was applied for all animals, including the KOs, after pharmacological treatment or chemogenetic treatment, so we believe our conclusions about the effects of these experimental conditions are accurate.

3. Fig 2: Is the correlation drop in a9/a9 KOs that is shown for P6-7 mice also present at other ages and do these KOs show a similar time course in peak amplitudes and event frequency?

Yes, the a9/a10 KOs have a correlation drop relative to Hets at P3-4 as well as P6-7. This data is shown in Supplementary Fig 3a-f and discussed at the top of page 7, around lines 172. We do not have a9/a10 KO data for other age groups. For the other response features (amplitude, frequency), KOs at P3-4, like P6-7, are not different from Hets (Supplementary Figure 3a-f). Comparing P3-4 and P6-7 KOs, the mean amplitudes and event frequencies appear to increase with age, but the differences are not statistically significant. We summarize this data and the relevant statistics here:

Age group	N (animals)	Amplitude (dF/F)	P value (t-test)	Event frequency (bands/min)	P value (t-test)
P3-P4 KO	6	0.1248	0.2497	59.18	0.1814
P6-P7 KO	14	0.1452		75.49	

4. A previous paper by Babola et al (Neuron, 2018) also reported some correlated activity bands in both IC hemispheres, which these authors attributed to a weak ipsilateral projection from the cochlear nucleus to the IC and the presence of commissural connections between both ICs. This interpretation was partly passed on the observation that there was a strong difference in the amplitudes of coordinated activity band between both ICs. Most importantly, even after unilateral cochlear ablation or unilateral pharmacological block of cochlear AMPA receptors, coordinated bilateral bands were still present, which indicates that one cochlear alone can drive both IC hemispheres. While these results do not exclude the possibility that MOC-mediated bilateral coordination exists, it indicates that the MOC is a contributor rather than the main driver of bilateral coordination. This is not made very clear in the discussion, title and introduction, the latter of which states that the MOC system is necessary for enforcing bilateral coupling. The authors could also address this by cutting the commissural connections and measuring the resulting correlation patterns.

As the reviewer rightfully points out, we should not have implied that the MOC system is the only one that contributes to bilateral coupling in the IC. We have modified the manuscript throughout, including the abstract (page 1 lines 21-27), introduction (page 2, around line 57) and discussion (page 12, around lines 347-354) to reflect this change. Our analysis shows that there are residual bilateral correlations at least in the medial IC (future low-frequency region) and, in some cases, more lateral regions at P11-P12 (Figure 1k), in a9/a10 KOs (Figure 2g), after acute pharmacology (Figure 3f, Supplementary Fig. 4e), and after chemogenetic silencing (Figure 4f). This suggests there are additional sources of bilateral coupling beyond the MOC system. We summarize all residual correlations and their significance in Supplementary Table 1 (copied here).

Conditions	Mean value	P value (one-sample t-test compared to 0)	Percentage correlation remaining compared to peak level in the same region at P6-7 or before manipulation
WT P11-P12 (average seed-based correlation) Figure 1k	0.1809	0.0003 ***	37.05% (62.95% reduction)
P6-7 a9a10 KO (low-frequency region) Figure 2g	0.1224	0.0387 *	21.35% (78.65% reduction)
P6-7 Acute Apamin (low-frequency region) Figure 3f	0.1752	0.0035 **	26.58% (63.42% reduction)
P6-7 Acute alpha-Bungarotoxin (averaged correlation) Supplementary Fig. 4e	0.2763	0.0086 **	45.95% (54.05% reduction)
P6-7 Inhibitory DREAD (averaged correlation) Figure 4f	0.3011	0.0008 ***	50.46% (49.54% reduction)

These results indicate that some bilateral coupling persists in the absence of MOC modulation, but the bulk of the bilateral correlations are attributable to modulation through the MOC.

Note that our previous study (Babola et al. Neuron 2018) developed the concept of a 'dominant' side as a potential indicator of unbalanced peripheral drive, but we didn't perform a detailed analysis of how big the 'dominance' was (there is always a "dominant" side as event amplitudes in the two hemispheres are impossible to be numerically identical) or explore the underlying biology. In particular, amplitude differences between the events in the two hemispheres may not necessarily mean the activity is only driven by one cochlea at a given time.

Although bilateral correlated bands persist after ablating/blocking one cochlea, the amplitude of ipsilateral activity (w.r.t. the blocked side) is substantially reduced. Such a reduction decreases the strength of bilateral correlations. On the other hand, we want to point out the conceptual difference of the underlying biology: when two cochleae are intact but the MOC modulation is absent, the two sides are generating and passing different/uncorrelated inputs to the central circuits simultaneously. For a downstream auditory nucleus that receives input from both sides, these two independent information streams might be confusing/work against each other. This is different from the case in which only one cochlea is left to drive downstream activity. In the latter case, there is no competing/confounding input from the ablated side. Therefore, the efferent system may be of particular importance in coordinating activity from two intact cochleae.

In summary, to clarify these differences, we have modified the manuscript throughout, including the abstract (page 1 lines 21-27), introduction (page 2, around line 57) and discussion (page 12, around lines 347-354).

For the last part of the reviewer comment, the reviewer suggests that we cut the IC commissure and then measure how much bilateral correlation persists without the commissure. We very much appreciate this idea and tried the experiment repeatedly (more than 10 pups), but it was extremely challenging due to: 1) Cutting the commissure along the midline between the two IC hemispheres almost unavoidably causes extensive damage to the medial collicular vein and/or transverse collicular artery, resulting in major hemorrhage and lethality in the neonates. Even minor vessel ruptures were problematic because they obscure the field of view for quantitative optical imaging; 2) Avoiding the midline vasculature by cutting the commissure laterally would damage one IC, rendering bilateral correlation analysis difficult to interpret.

4. MOC synapses inhibit inner hair cells. Would it not be expected that activation of MOC would decrease the peak amplitude of IC bands or decrease their frequency, which however, was not the case (Fig 4h, i). Similarly, previous recordings in prehearing alpha9 KO mice showed a significant shortening of activity burst in these mice (Clause et al., 2014). Since the length of these burst is expected to influence the duration of IC bands one would expect a decrease of and duration in the a9/a10 KO mice. The authors analyzed duration in WT mice but not KO mice but it would be interesting and informative to know whether band duration is reduced in the KO and CNO treated DREADDs mice. The authors already have the data, so the analysis should be included in the paper.

Thank you for the suggestion, we have included the suggested analysis as Supplementary Fig. 3j (KO) and Supplementary Figure 5d (CNO treated DREADDs animals) and added text to the manuscript to describe these results (page 6, around line 167; page 8, around line 220). Based on the analysis, we did not see a significant change in average band durations in KO mice or CNO treated mice relative to controls.

Note that we did observe a mild but significant increase in the peak amplitude and event frequency in a9/a10 KO mice relative to Hets (Fig. 2), but the effect on bilateral correlations was much more profound. It is a little unexpected that we did not observe a difference in event frequency, amplitude or duration upon application of inhibitory or excitatory dreads (Fig. 4). We think several factors might mask these effects: 1) The magnitude of efferent modulation on IHCs is more pronounced when measuring correlations than amplitude, frequency or duration. Thus, the incomplete effects of DREADDs may be easier to observe on correlations than the other measures. 2) In the Clause et al. (2014) study, the authors also reported *unchanged* “overall firing rates” and “bursts/min” in MNTB neurons from KO mice, similar to the event frequency measurement we use. It is also possible that homeostatic mechanisms balance out these effects in the inferior colliculus relative to the MNTB. 3) While Clause et al. (2014) reported a shorter firing burst duration in KOs, they also observed a higher firing rate during a burst. These two changes may offset each other at the mesoscopic scale reported here when imaging the IC. Ca²⁺ signals (based on GCaMP6s) have very slow dynamics, with the half decay time of GCaMP6s ranging from one to several seconds (in response to one to one hundred action potentials, please see Figure 1f in the original GCaMP6s paper <https://www.ncbi.nlm.nih.gov/pmc/articles/PMC3777791/>). The intense bursting spike pattern (Clause et al., 2014) recorded in the a9 KOs would be expected to prolong calcium signal dynamics compared to WT mice, potentially offsetting the shortened burst duration. 4) Electrophysiology results acquired at the single cell level in vitro are quite different than global properties of hundreds of mesoscopic spontaneous bands measured in vivo. Even a single spontaneous band reflects collective activity of tens of thousands of neurons. Therefore, dynamics of mesoscopic spontaneous bands might not be sensitive enough to reflect changes at the single neuron level. In general, we think if there is something significant at the wide/coarse scale, there is highly likely some changes at a finer scale while the opposite is not necessarily true. We added an explanation of these differences to the discussion section (paragraph at the top of page 13, around lines 364-382) in the manuscript.

5. It is not quite clear why alpha-Bungarotoxin was applied to both cochleae. Should unilateral blockage not be sufficient for decorrelation?

Unilateral application of alpha-Bungarotoxin/apamin would block MOC modulation on the manipulated side, while MOC neurons receiving inputs from the manipulated side but modulating the other/untouched side could still function. Based on pilot experiments, we observed a more complete effect when applying drugs to both cochleae, so this is the approach we pursued.

6. It is becoming increasingly clear that CNO can have potential off-target effects. Thus, it is necessary to present the effect or lack thereof of administering CNO in WT animals.

Thank you for suggesting this control experiment. We have included it as Supplementary Figure 5e-h and added text to lines 223-224 (page 8).

Based on the results, CNO in WT animals has no effect on bilateral correlations, confirming the chemogenetic effect of CNO in experimental animals is due specifically to the expression of DREADDs.

Fig 5d is not very informative and can be deleted to make space enlarge 5C.

Thank you for the suggestion, we have modified Fig. 5 as requested.

7. Looking at the traces of sound evoke activity in IC, it seems that the KO mice had a less bilaterally correlated activity in the IC than the Het mice. Given that there was free-field stimulation this is a bit surprising, if the images shown are representative.

We do sometimes see “unbalanced” evoked activity in IC, but not only in the KO mice. This phenomenon is most obvious for responses evoked by weak stimuli just above threshold (for example in Fig 5c. P14.0 a9 Het/ a10 Het, 8 & 11.3 kHz/ 32 dB). The effect could potentially be stronger in the KO mice, and it might be an interesting point to investigate in future studies.

8. The ABR traces look atypical compared to the waveforms that are generally published. This is especially obvious for the trace from the P60 animal that in which waves ride on each other instead of having propounded negative peaks.

This is mostly due to the setting we used for bandpass. We set the filter at a wider range (50Hz and 3 kHz), which allows us to also record low-frequency components. Although this setting might permit some forms of noise, such as breathing rhythms to get through, we would like to preserve these components because they potentially contain LFP responses from the inferior colliculus (the “p0 wave”, see Figure 1c-d in Land et al., 2016 <https://www.sciencedirect.com/science/article/pii/S0378595516303288>, where the authors compare ABR results with different bandpass widths). These signals are important for our future research. The reviewer might refer to “typical” waveforms given by the 300Hz-3kHz bandpass filter. We replotted Figure 5a with a more representative trace from a different mature wildtype animal recorded at a similar time with the same setting (50Hz-3kHz bandpass filter). This example may be more typical of what the reviewer expects. We also specify our bandpass filter width in Methods (around line 840).

9. Line 137: Local connectivity. The authors speculate the existence of developmental changes in local connectivity. The authors may be referred to Sturm et al., 2014, who investigated exactly this issue.

Thank you for pointing this out, we added this context and reference to our discussion (page 15, around lines 473-476).

10. Often, the labeling for the look-up tables and significance bars and so small that it is undecipherable. Also, even if a simple computer screen shot is show in the figure, axes should be labelled.

Thank you for the suggestion, we have adjusted all figures and added labels. All figure legends can also be seen in the main manuscript document.

11. Line 163 – “a9/a10” Het should read “a9 KO/a10 het”?

Yes, thank you for catching this! We have changed the text.

Reviewer #2 (Remarks to the Author):

The paper by Wang et al. charts the profile of bilateral coupling in the developing inferior colliculus (IC). The authors demonstrate that correlated spontaneous activity across the two hemispheres peaks several days prior to hearing onset, and that coupling is dependent on cholinergic feedback to the cochlea. Disruption of this pathway is associated with a loss of auditory sensitivity at hearing onset. This is a very interesting study, and I particularly admire the range of complementary approaches that have been used to demonstrate how altering cholinergic feedback to cochlea influences cross-hemisphere correlations at the level of IC - genetic knockouts, pharmacology and chemogenetics. There are some aspects that require further clarification:

1. There is always a concern that changes in correlation strength are purely the result of changes in the number and/or amplitude of events. Here, sub-significant changes (or barely significant changes, i.e. Fig2D) in the frequency and amplitude of spontaneous calcium events could be leading to significant changes in correlation. Could the authors present a quantitative refutation of this idea? For example, they could break recording sessions down into epochs and then (across all animals) measure the relationship between event frequency (in these epochs) with the correlation values (likewise with the event amplitude) and, hopefully, show no relationship exists.

Thank you for pointing out that changes in amplitude/event frequencies, if large enough, can affect measured correlations. However, we do not believe that changes in amplitude/event frequencies are responsible for the changes in correlation we observed for the following reasons:

For bilateral correlations vs. amplitudes: Below we plot averaged seed-based correlations against amplitudes for all P0-P12 animals in Figure 1 (N = 49). With linear regression, it shows F-statistic vs. constant model: 0.0536, p-value = 0.818, suggesting no relationship between these two quantities.

For bilateral correlations vs. event frequencies: Below we plot event bilateral correlations vs. frequencies for all P0-P12 animals in Figure 1 (N = 49). With linear regression, it shows F-statistic vs. constant model: 9.49, p-value = 0.00344, suggesting a negative correlation between these two quantities across ages.

However, the negative relationship does not indicate increasing event frequency is causing bilateral correlations to decrease. Actually, if we look at Figure 1 d-e, the amplitude and event frequency of age group P3-4 & P11-12 are both significantly higher than those of age group P0-1. However, the P3-4 age group has higher seed-based correlations while the P11-12 group has lower ($p = 0.001$, significance bar not shown on the figure due to space limit), compared to the P0-1 group (Figure 1k). This suggests that changes of event frequency and amplitude in one direction led to changes of correlations in different directions. In fact, the negative relationship

above is mostly driven by low bilateral correlation values of the P11-P12 age group. When we plot the event frequency against bilateral correlation in P0-P9 animals, we have F-statistic vs. constant model: 0.99, p-value = 0.326 (no linear relationship). See below:

On the other hand, In Figure 3 and 4, event frequency and amplitude remain at a similar level before/after different manipulations (pharmacology or chemogenetics), while bilateral correlations can either decrease (Figure 3 and Figure 4 d-f, inhibitory DREADD part) or increase significantly (Figure 4 j-l, excitatory DREADD part). All these results mentioned suggest that, within the range of this study, activity level and correlations are decoupled.

2. Why does altering feedback, via activation of excitatory/inhibitory DREADDs, not alter event frequency or event amplitude? If activation of the MOC, via spontaneous cochlea activation, is enough to drive synchronized activation of both hemispheres of the IC then why is fundamentally altering general cochlea activity using chemogenetics not sufficient to alter event amplitude or frequency at all?

This question is similar to Reviewer #1, Question 4. Please also refer back to that question and answer. We added an explanation of these differences to the discussion section (paragraph at the top of page 13, around lines 364-382) in the manuscript.

In summary, eliminating efferent feedback likely has complex effects on spontaneous activity, some of which may not be obvious with wide field in vivo Ca²⁺ imaging of the IC. With in vitro techniques. Clause et al. (Figure 1, Clause et al., 2014, <https://www.ncbi.nlm.nih.gov/pmc/articles/PMC4052368/>) reported *no difference* in the burst frequency or firing rate in a9 KO mice, despite the absence of feedback inhibition. Clause et al. (2014) *did observe* a decrease in burst duration, but a commensurate increase in the number of spikes per burst. These compensatory effects might mask any changes when imaging large-scale mesoscopic activity patterns, as each spontaneous band manifests collective activity from tens of thousands of neurons, and the GCaMP6s signals are slow and accumulative. Also, although activity in the IC may inherit many properties from the periphery, as these inputs propagate up the ascending auditory system there are certainly going to be some changes in activity dynamics.

3. The movies look as though there are delays between left and right (or sometimes that intensity increase once both are active). Did the authors look for temporal delays between hemispheres to understand to what extent cross-talk influenced spontaneous events? – This might be important as I think they suggest the coupling happens at the input level not the IC level?

Thank you for the suggestion! We analyzed timing differences between pairs of temporally neighboring spontaneous events in left and right IC hemispheres in P6-7 control animals (see the cumulative distribution & histogram above). Most of the time, when there is an event in one hemisphere, there is also a temporally neighboring event identified in the opposite hemisphere (40% within 0-1 frame or 100 msec, 58% within 5 frames or 500 msec). We should note that the timing differences here were computed in terms of “peaks” of activity, not the onset of activity, further confounding interpretation of temporal delays. Peaks (maxima) of activity can be easily

picked identified quantitatively, but it is very hard to infer or even appropriately define the time of onset of an event, for several reasons. 1) There is no clear external reference to signal onsets of spontaneous events. 2) The temporal dynamics of GCaMP is thought to be dependent on the amplitude (strength) of activity. 3) The mesoscopic calcium activity we image reflects collective activity from thousands of neurons, introducing further heterogeneity. 4) It is challenging to tease apart whether a pair of peaks in two IC hemispheres recorded within a small temporal window should be attributed to the same peripheral spontaneous event (with a delay) or different events that happen to occur close together in time. 5) The timescale here is slow (movies acquired at 10Hz, GCaMP6s decay time constant ~ 1 s), thus a single-frame delay indicates a 100ms temporal difference. This resolution is not enough to infer much about fast dynamics. In summary, it is unclear to what extent temporal differences of optical salience (identified quantitatively through an analysis algorithm or the human eye) accurately reflects temporal differences of underlying biological events at fast time scales. Without addressing these caveats, it will be hard to interpret any potential results given the imaging resolution we have here. The advent of much faster dynamical activity indicators, potentially voltage sensitive dyes, will be necessary for this higher-level analysis.

4. “In addition, we noticed a tonotopic difference in bilateral correlations using SbBCs, with low frequency regions exhibiting stronger correlations than those of high frequency regions at P6-7 (Supplementary Data Fig. 2d), suggesting that coupling strengths can vary by tonotopic location.” – Could this also be due to a better field of view of these areas?

We think this is unlikely because: 1) The optical aberrations are modest, as both the low and high frequency regions are not near the edge of the microscope’s field of view. 2) It is possible that the curvature of the IC could impact the spatial shape of the activity patterns, but they should not impact the temporal features of the activity. Although we cannot rule out the possibility that surface curvatures might slightly affect the observed amplitudes of spontaneous events, we showed above (First figure in our answer to Question 1) that the amplitude of spontaneous events is not related to the strength of bilateral correlations. Thus, it is unlikely that the observed tonotopic differences in bilateral correlations is an optical artifact of the imaging field of view.

5. 236-237: It would be useful to include an estimate of how much thresholds changed (even if just a mean value across frequency).

Thank you for the suggestion, we added this analysis to the results section (page 9, around line 262). The average difference (across all frequencies) was around 20dB.

6. 242-243: “and there was little difference in responses in $\alpha 9/\alpha 10$ knockout and control animals, except at the lowest frequency (Fig. 5a)” - Here the authors use a t-test with multiple

comparisons correction. Why not an overall statistic which would be more sensitive to general differences? If they are all slightly different it suggests that overall, they might be significantly different.

Thank you for the suggestion! We substituted the multiple t-tests with two-way ANOVA tests in the manuscript, Figure 5 and Figure 5 legend. The result indicates that the genetic factor (knockouts vs. controls) significantly elevated the auditory sensitivity in general (across the frequency range) for the wide-field calcium imaging dataset/IC activity (Fig. 5c, $F = 31.98$, **** $p < 0.0001$) but the difference did not reach statistical significance for the ABR dataset (Fig. 5b, $F = 1.409$, $p = 0.0604$).

7. 439-440 “that is critical to the development of normal auditory function” – the data show that hearing thresholds are elevated very soon after hearing onset (P14). However, it is not known whether hearing thresholds can subsequently recover. The text should be amended to reflect this uncertainty.

Thank you for the suggestion! We added this point to our discussion on auditory thresholds (page 14; around lines 401-402).

8.

Figure 1 – panel c is mislabelled; panel g requires time calibration bar.

Figure 4 – panels a, g – ‘Imged’ > imaged

Figure 5 – panel e – legend typo

line 381 – “lowers” = lower

line 393 – “Frist,” = first

Thank you for catching these errors! We fixed them in the revised manuscript.

Reviewers' Comments:

Reviewer #1:

Remarks to the Author:

The authors addressed all of my critiques and I have no further comments.

Reviewer #2:

Remarks to the Author:

This is an interesting study and the authors have addressed my concerns. Thank you.